# Does the rotational direction of a wind turbine impact the wake in a stably stratified atmospheric boundary layer?

Antonia Englberger[1], Andreas Dörnbrack[1], and Julie K. Lundquist[2,3]

[1]German Aerospace Center, Institute of Atmospheric Physics, Oberpfaffenhofen, Germany
[2]Department of Atmospheric and Oceanic Sciences, University of Colorado Boulder, Boulder, USA
[3]National Renewable Energy Laboratory, Golden, Colorado, USA

**Correspondence:** Antonia Englberger (antonia.englberger@dlr.de)

**Abstract.** Stably stratified atmospheric boundary layers are often characterized by a veering wind profile, in which the wind direction changes clockwise with height in the Northern Hemisphere. Wind-turbine wakes respond to this veer in the incoming wind by stretching from a circular shape into an ellipsoid. We investigate the relationship between this stretching and the direction of the turbine rotation by means of large-eddy simulations. Clockwise rotating, counterclockwise rotating, and non-rotating actuator disc turbines are embedded in wind fields of a precursor simulation with no wind veer and in wind fields with a Northern Hemispheric Ekman spiral, resulting in six combinations of rotor rotation and inflow wind condition. The wake strength, extension, width and deflection depend on the interaction of the meridional component of Ekman spiral with the rotational direction of the actuator disc, whereas the direction of the disc rotation only marginally modifies the wake if no veer is present. The differences result from the amplification or weakening/reversion of the spanwise and the vertical wind components due to the effect of the superposed disc rotation. They are also present in the streamwise wind component of the wake and in the total turbulence intensity. In the case of a counterclockwise rotating actuator disc, the spanwise and vertical wind components increase directly behind the rotor resulting in the same rotational direction in the whole wake while its strength decreases downwind. In the case of a clockwise rotating actuator disc, however, the spanwise and vertical wind components of the near wake are weakened or even reversed in comparison to the inflow. This weakening/reversion results in a downwind increase of the strength of the flow rotation in the wake or even a different rotational direction in the near wake in comparison to the far wake. The physical mechanism responsible for this difference can be explained by a simple linear superposition of a veering inflow with a Rankine vortex.

## 1 Introduction

In wind energy science, the engineering system of a wind turbine interacts with the geophysical system of the atmospheric boundary layer (ABL). The canonical ABL over land experiences a diurnal cycle in prevailing wind and turbulence conditions (Stull, 1988). The diurnal cycle is driven by shortwave heating during the day and radiative cooling at night. Shortwave heating of the surface triggers convective turbulence that mixes throughout the boundary layer, resulting in a well-mixed convective boundary layer (CBL) during day with little vertical wind shear (change of wind speed with height) and high levels of turbulence. The blades of a wind turbine in a CBL are therefore often exposed to the same wind speed and wind direction at each possible blade position. At night, radiative cooling of the surface leads to a decay of the convective turbulence, resulting in a stable boundary layer (SBL), with low turbulence levels and highly sheared wind profiles. The interaction between the Coriolis force and friction in the boundary layer will cause winds to rotate with height. This veering of the wind through the boundary layer is described by the Ekman spiral. In the Northern Hemisphere, winds will rotate in the clockwise direction with height (i.e. westerly near the surface and northerly aloft), while in the Southern Hemisphere, winds will rotate in the counterclockwise direction with height (i.e. westerly near the surface and southerly aloft). Therefore, the nighttime wind system in the Northern Hemispheric mid-latitudes is typically characterised by a veering wind (a wind that rotates in a clockwise direction with height) and a pronounced vertical wind shear (Walter et al., 2009). Of course, synoptic events such as frontal passages or topographically-driven phenomena such as drainage flows may modify this typical background veer. The blades of a wind turbine in an SBL therefore do not interact with a uniform flow field in all heights as during day, but with an increasing wind speed with height and also a wind direction change with height.

The stability-dependent wind and turbulence conditions of a wind-turbine inflow determine the entrainment of energy and momentum into the wake region and the resulting wake structure, with fast eroding wakes in convective conditions and wakes persisting farther downwind in stably stratified conditions. This stability dependence has been investigated using large-eddy simulation (LES) for the SBL (Aitken et al., 2014; Bhaganagar and Debnath, 2014, 2015; Dörenkämper et al., 2015), the CBL (Mirocha et al., 2014), both of them (Abkar and Porté-Agel, 2014; Vollmer et al., 2017), or the complete diurnal cycle (Abkar et al., 2016; Englberger and Dörnbrack, 2018). This stability dependence has also been observed in field campaigns (Iungo et al., 2013; Bodini et al., 2017).

All modern multi-MW wind turbines have blades that rotate clockwise when looking downwind at the turbine. Historically, traditional grain-grinding wind mills turned counter-clockwise for ease of manufacturing by right-handed technicians. By market happenstances, described well in Maegaard et al. (2013), the companies building blades in the 1970's evolved from having a mix of clockwise and counterclockwise rotating blades into a market domination of clockwise rotating blades. These clockwise rotating blades exhibit a counterclockwise rotating wake and vice versa (Zhang et al., 2012): due to aerodynamics and design of the wind turbine blades, the flow moves the blades in one direction and is deflected away from them in the

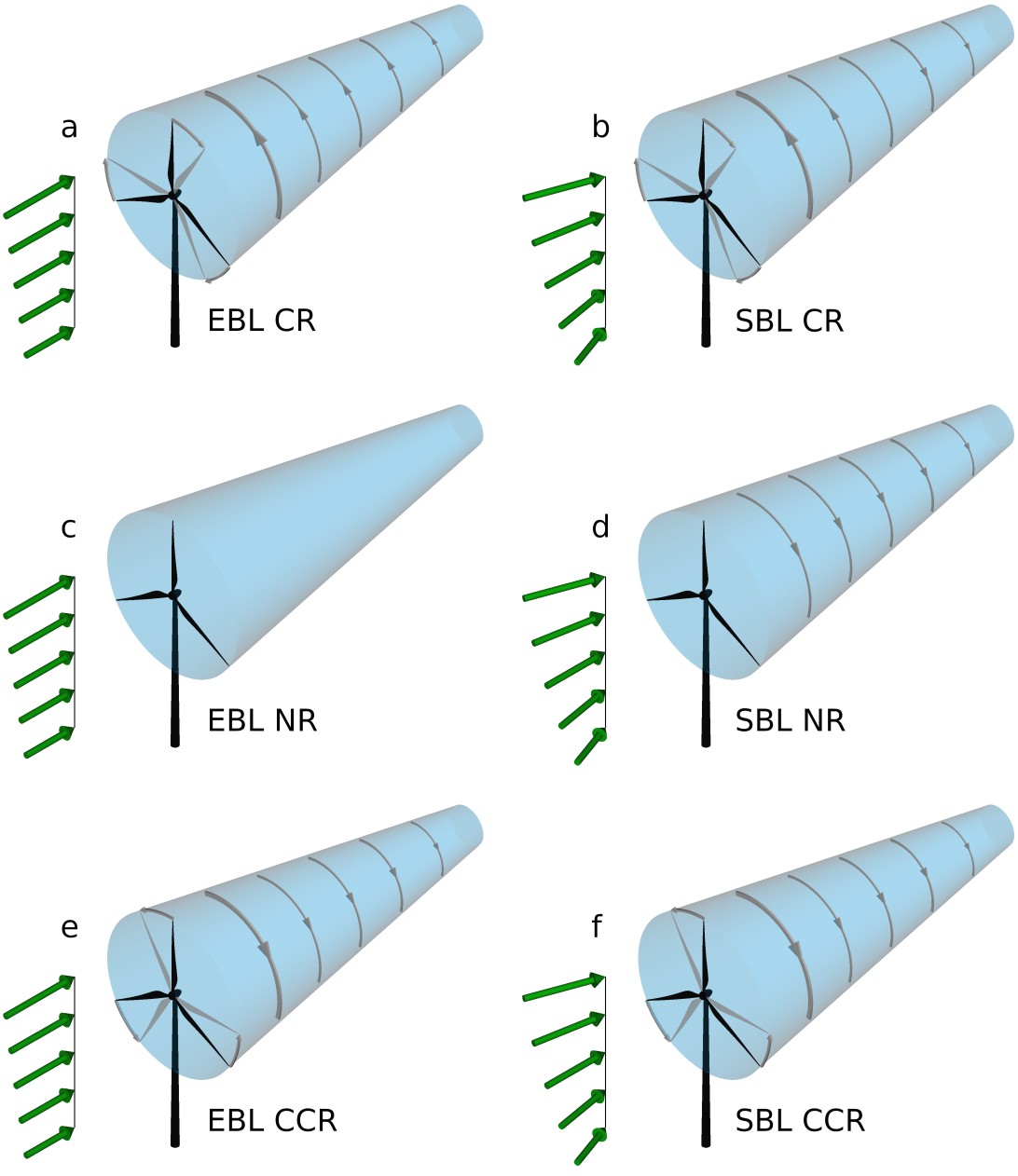

**Figure 1.** Schematic illustration of the rotational direction of the wake for the cases: No wind veer with clockwise blade rotation (EBL CR) in (a), wind veer with clockwise blade rotation (SBL CR) in (b), no wind veer with no blade rotation (EBL NR) in (c), wind veer with no blade rotation (SBL NR) in (d), no wind veer with counterclockwise blade rotation (EBL CCR) in (e), wind veer with counterclockwise blade rotation (SBL CCR) in (f).

opposite direction. Studies investigating the effect of wind turbine rotors rotating in opposite directions, especially for wind farm optimization, show that the rotational direction has an impact on the wake structure of a wind turbine and therefore on the performance of a downwind turbine (Vermeer et al., 2003; Shen et al., 2007; Sanderse, 2009; Kumar et al., 2013; Hu et al., 2013; Yuan et al., 2014; Mühle et al., 2017). Further, in simulations, representing an array of wind turbines with a second row
rotating opposite to the rotation of the first row, an increase in productivity was found in comparison to the co-rotating pair of turbines. This improvement is related to the different direction of the angular component in the wake and therefore the change in the angle of attack on the downwind turbine (Mühle et al., 2017).

However, these studies were performed for wind conditions without significant vertical wind shear and without wind veer in the rotor altitudes. Vertical wind shear and wind veer impact wake characteristics (Abkar and Porté-Agel, 2016; Abkar
et al., 2016; Vollmer et al., 2017; Bromm et al., 2017; Churchfield and Sirnivas, 2018; Englberger and Dörnbrack, 2018), power production (Gomez and Lundquist, 2020), as well as turbine loads (Kapoor et al., 2019). Because both shear and veer influence the wake, their interaction in combination with different rotational directions might impact wake characteristics in stably stratified regimes. This possible impact is investigated in this study in detail, aiming to answer the question:

Does the rotational direction of a wind turbine impact the wake in a stably stratified atmospheric boundary layer?

We investigate the relationship between the upstream wind profile, the direction of the turbine rotation, and the wake by LES. Clockwise-rotating, counterclockwise-rotating and non-rotating actuator discs are embedded in two different atmospheric regimes taken from a diurnal cycle simulation from Englberger and Dörnbrack (2018). The evening boundary layer (EBL) is characterized by a small amount of vertical wind shear with no wind veer. The SBL is characterized by a significant amount of vertical wind shear and wind veer, representing the Northern Hemispheric Ekman spiral. The simulations represent the six
combinations of rotor rotation and wind conditions shown in Fig. 1. To our knowledge, this study is the first investigation of the dependence of wake characteristics on the rotational direction of a rotor and wind veer in a stably stratified ABL.

The paper is organised as follows. The numerical model EULAG, the metrics, and the wind-turbine simulation setup are described in Sect. 2. The analysis of a rotating system under veering inflow is presented in Sect. 3, and the wake characteristics of a rotating wind-turbine rotor under veering inflow is presented in Sect. 4. A comparison of a rotating wind turbine under
veering inflow to analysis predictions is given in Sect. 5. A conclusion follows in Sect. 6.

## 2    Numerical Model Framework

### 2.1    The Numerical Model EULAG

The dry ABL flow through a wind turbine is simulated with the multiscale geophysical flow solver EULAG (Prusa et al., 2008; Englberger and Dörnbrack, 2017). The acronym EULAG refers to the ability to solve the equations of motion either
in an EUlerian (flux form) (Smolarkiewicz and Margolin, 1993) or in a semi-LAGrangian (advective form) (Smolarkiewicz and Pudykiewicz, 1992) mode. The geophysical flow solver EULAG is at least of second-order accurate in time and space (Smolarkiewicz and Margolin, 1998) and well-suited for massively-parallel computations (Prusa et al., 2008). A comprehensive

description and discussion of the geophysical flow solver EULAG can be found in Smolarkiewicz and Margolin (1998) and Prusa et al. (2008).

For the numerical simulations, the Boussinesq equations for a flow with constant density $\rho_0 = 1.1$ kg m$^{-3}$ are solved for the Cartesian velocity components $\mathbf{v} = (u, v, w)$ and for the potential temperature perturbations $\Theta^{'} = \Theta - \Theta_e$ (Smolarkiewicz et al., 2007),

$$\frac{d\mathbf{v}}{dt} = -G\boldsymbol{\nabla}\left(\frac{p^{'}}{\rho_0}\right) + \mathbf{g}\frac{\Theta^{'}}{\Theta_0} + \boldsymbol{\mathcal{V}} + \mathbf{M} - 2\,\Omega_C\left(\mathbf{v} - \mathbf{v}_e\right) + \boldsymbol{\beta_{\mathbf{v}}}\frac{\mathbf{F}_{WT}}{\rho_0}, \tag{1}$$

$$\frac{d\Theta^{'}}{dt} = -\mathbf{v}\nabla\Theta_e + \mathcal{H}, \tag{2}$$

$$\boldsymbol{\nabla} \cdot (\rho_0 \mathbf{v}) = 0, \tag{3}$$

where $\Theta_0$ represents the constant reference value. Height dependent states $\psi_e(z) = (u_e(z), v_e(z), w_e(z), \Theta_e(z))$ enter Eqs. 1 - 3 in the buoyancy term, the Coriolis term, and as boundary conditions. These background states correspond to the ambient and environmental states. Initial conditions are provided for $u$, $v$, $w$, and $\Theta^{'}$ in $\psi = (u, v, w, \Theta^{'})$. In Eqs. (1), (2) and (3), $d/dt$, $\boldsymbol{\nabla}$ and $\boldsymbol{\nabla} \cdot$ represent the total derivative, the gradient and the divergence, respectively. The quantity $p^{'}$ represents the pressure perturbation with respect to the background state and $\mathbf{g}$ the vector of acceleration due to gravity. The factor $G$ represents geometric terms that result from the general, time-dependent coordinate transformation (Wedi and Smolarkiewicz, 2004; Smolarkiewicz and Prusa, 2005; Prusa et al., 2008; Kühnlein et al., 2012). The subgrid-scale terms $\boldsymbol{\mathcal{V}}$ and $\mathcal{H}$ symbolise viscous dissipation of momentum and diffusion of heat and $\mathbf{M}$ denotes the inertial forces of coordinate-dependent metric accelerations. The Coriolis force is represented by the angular velocity vector $\Omega_C$ of the Earth's rotation. All following simulations are performed with a TKE closure (Schmidt and Schumann, 1989; Margolin et al., 1999).

The axial $\mathbf{F}_x$ and tangential $\mathbf{F}_\Theta$ turbine-induced forces ($\mathbf{F}_{WT} = \mathbf{F}_x + \mathbf{F}_\Theta$) in Eq. (1) are parametrized with the blade element momentum (BEM) method as a rotating actuator disc without and with rotation in clockwise and counterclockwise direction ($\boldsymbol{\beta_v} \in \{0, -1, 1\}$), including a nacelle and excluding the tower. We do not simulate the rotor rotation directly, instead, we exert the rotor forces directly on the velocity fields (Eq. (1)). As a clockwise wake rotation is initiated by a counterclockwise blade rotation, due to conservation of angular momentum (e.g. described in Zhang et al. (2012)), we define a common clockwise rotor rotation as counterclockwise wake rotation with $\beta_v = 1$ and $\beta_w = -1$, a counterclockwise rotor rotation as clockwise wake rotation defined by $\beta_v = -1$ and $\beta_w = 1$, and no rotation is simulated by $\beta_v = 0$ and $\beta_w = 0$, with $\beta_u = 1$ in each case.

The BEM method accounts for local blade characteristics, as it enables calculation of the steady loads as well as the thrust and the power for different wind speeds, rotational speeds, and pitch angles of the blades. For the airfoil data, the 10 MW reference wind turbine from DTU (Bak et al., 2013) is applied, whereas the radius of the rotor as well as the chord length of the blades are scaled to a rotor with a diameter of 100 m. A more detailed description of the wind-turbine parametrization and the applied smearing of the forces, as well as all values used in the wind-turbine parametrization are given in Englberger and Dörnbrack (2017, parametrization B).

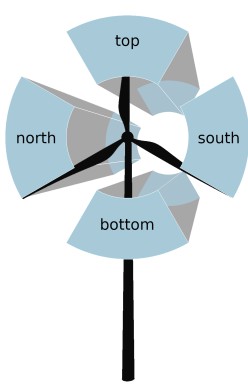

**Figure 2.** Schematic illustration of the top sector, the bottom sector, as well as the southern and northern sectors, defined from a view looking downwind from west to east towards the wind turbine on the disc. All sectors are $90°$ with $25\,\text{m} \leq \text{r} \leq 50\,\text{m}$.

## 2.2 Metrics

For the investigation of the dependence of wake characteristics on the wake rotation and wind veer, the following characteristics are calculated from the simulation results: the spatial distribution of the time-averaged discrete streamwise velocity $\overline{u_{i,j,k}}$, the time-averaged discrete spanwise velocity $\overline{v_{i,j,k}}$, the time-averaged discrete vertical velocity $\overline{w_{i,j,k}}$, the streamwise velocity deficit

$$VD_{i,j,k} \equiv \frac{\overline{u_{1,j,k}} - \overline{u_{i,j,k}}}{\overline{u_{1,j,k}}}, \tag{4}$$

the streamwise ($x$), spanwise ($y$) and vertical ($z$) turbulence intensity, e.g. in $x$ direction

$$TI_{x_{i,j,k}} = \frac{\sigma_{u_{i,j,k}}}{\overline{u_{i,j,k_h}}}, \tag{5}$$

with $\sigma_{u_{i,j,k}} = \sqrt{\overline{u'^2_{i,j,k}}}$ and $u'_{i,j,k} = u_{i,j,k} - \overline{u_{i,j,k}}$. Similar formulas apply for $TI_y$ and $TI_z$ in the $y$, and $z$ directions respectively. The total turbulence intensity is

$$TI_{i,j,k} = \sqrt{\frac{1}{3}(TI^2_{x_{i,j,k}} + TI^2_{y_{i,j,k}} + TI^2_{z_{i,j,k}})}. \tag{6}$$

The indices of the grid points are denoted by $i = 1 \dots n$, $j = 1 \dots m$ and $k = 1 \dots l$ in the $x$, $y$ and $z$ directions, respectively. These characteristics are averaged over the last 50 min of the corresponding 1 h wind-turbine simulation. The temporal average is calculated online in the numerical model and updated at every timestep according to the method of Fröhlich (2006, Eq. 9.1).

In the following, the quantities $\overline{u_{i,j,k}}$, $\overline{v_{i,j,k}}$, and $\overline{w_{i,j,k}}$ are evaluated and discussed downwind of the wind turbine up to 20 D. Further, the rotor area is divided into four sections of $90°$, as shown in Fig. 2, including all grid points with a distance $r$ from the rotor center $R/2 \leq \text{r} \leq \text{R}$, referred to hereafter as top sector, bottom sector, northern sector and southern sector, looking from upwind ($i = 1$) downwind toward the actuator. In this work, we refer to the left sector as northern sector and to the right

sector as southern sector, defined from a view looking downwind from west to east towards the wind turbine on the disc. This naming results from a zonal wind from west to east representing the streamwise flow and a meridional wind from south to north representing the spanwise flow through the wind turbine in the analyzed regimes of this work. However, the results of this work are valid independent of the wind direction of the streamwise and spanwise flow through the wind turbine. In any considered case, a left sector replaces the northern sector and a right sector replaces the southern sector.

## 2.3  Setup of the Wind-Turbine Simulations

In this work, a wind turbine is placed in two different regimes characteristic of different parts of the diurnal cycle: the EBL and the SBL. Both regimes develop from a diurnal cycle precursor simulation over a homogeneous surface with periodic horizontal boundary conditions (Englberger and Dörnbrack, 2018). The simulation includes 512 x 512 grid points in the horizontal direction with a resolution of 5 m. The vertical resolution is also 5 m in the lowest 200 m, stretching up to the top of the domain at 2 km. The precursor simulation is initialized with a geostrophic wind of 10 m s$^{-1}$ as zonal (west-east) wind and no meridional (south-north) and vertical wind. The initial potential temperature is set to a constant value of 300 K up to 1 km, increasing above with a lapse rate of 10 K km$^{-1}$. A sensible heat flux forces the diurnal cycle with a maximum of 140 W m$^{-2}$ during day and a minimum of -10 W m$^{-2}$ during night. The diurnal cycle simulation is initialized at 0000 UTC. For the EBL regime, a 1 h time period from 1800 UTC to 1900 UTC is extracted. During this time period the surface fluxes changes sign. For the SBL regime, the selected 1 h time period starts at 0000 UTC (24 h after initialization). The horizontally averaged zonal ($u$) and meridional ($v$) velocity profiles, the vertical potential temperature ($\Theta$) profile, the total turbulence intensity ($TI$) (Eq. 6) and the turbulence intensities in x y, and z direction ($TI_x$, $TI_y$, $TI_z$) (Eq. 5) of the beginning of the corresponding atmospheric regime are presented in Fig. 3.

Three factors distinguish the EBL and the SBL from each other: First, the meridional wind $v$ is zero with $\frac{\partial v}{\partial z} = 0$ in the EBL, whereas the $v$-component in the SBL is larger than zero with $\frac{\partial v}{\partial z} \neq 0$ for $z < 100$ m (Fig. 3(a)). This veering wind at night is the main difference to the non-veered wind in the EBL. The resulting wind direction change with height in the SBL is an effect of the Coriolis force acting on the velocity components and friction due to the surface in combination with low levels of turbulence due to radiative cooling of the surface at night. The veering wind profiles correspond to the Ekman spiral with

$$\gamma = \sqrt{\frac{f}{2\kappa}}$$

for a Coriolis parameter f $= 1.0 \times 10^{-4}$ s$^{-1}$, and an eddy viscosity coefficient $\kappa = 0.05$ m$^2$ s$^{-1}$, which is an appropriate value for an SBL regime (Yamada and Mellor, 1975). This results in the zonal and meridional wind profiles representing the Ekman spiral

$$u_{Ekman}(z) = u_g \cdot (1 - exp(-z\gamma) cos(z\gamma)), \tag{7}$$

$$v_{Ekman}(z) = u_g \cdot (exp(-z\gamma) sin(z\gamma)), \tag{8}$$

following Stull (1988), with a geostrophic wind $u_g = 10$ m s$^{-1}$ in Fig. 3(a).

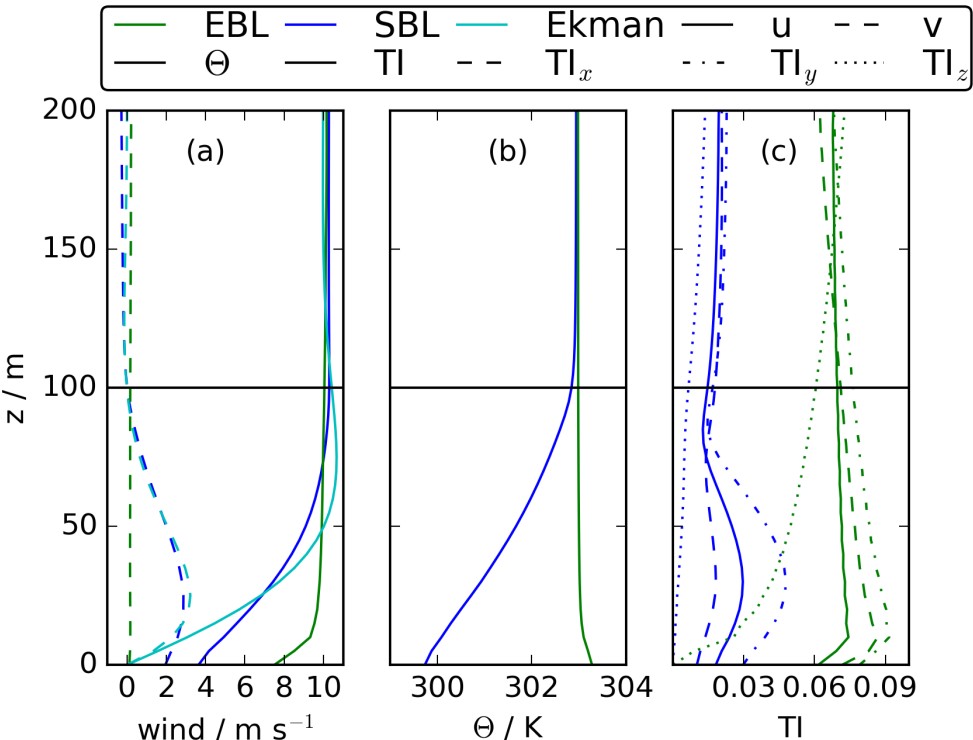

**Figure 3.** Vertical profiles of the horizontal wind components $u$ and $v$ in (a), the potential temperature $\Theta$ in (b), and the total $TI$, streamwise $TI_x$, spanwise $TI_y$, and vertical turbulence intensity $TI_z$ in (c). The profiles represent the horizontal averages of the EBL and the SBL-regimes in the precursor diurnal cycle simulation.

Second, the potential temperature profiles differ between the EBL and the SBL (Fig. 3(b)). The sensible heat flux of -10 W m$^{-2}$ during night leads to a maximum cooling close to the surface in the SBL regime, whereas $\frac{\partial \Theta}{\partial z} \approx 0$ in the EBL regime when the surface fluxes changes sign and are therefore very small.

Finally, the EBL and the SBL also differ in the amount of turbulence in the atmosphere. Shortwave heating of the surface during day triggers convective turbulence. The EBL still experiences well mixed turbulence, resulting in higher levels of turbulence intensity in comparison to the SBL, with a minor variation in height of $TI$, $TI_x$, and $TI_y$ (Fig. 3(c)). At night, radiative cooling of the surface results in negative buoyancy which damps turbulence. Therefore, the total turbulence intensity as well as all three individual components of $TI$ are smaller in the SBL regime in comparison to the EBL regime (Fig. 3(c)). A more detailed description of the diurnal cycle parameters from Fig. 3 is presented in Englberger and Dörnbrack (2018).

The EBL regime and the SBL regime represent the two atmospheric regimes in the corresponding wind-turbine simulations. The wind-turbine simulations are performed with periodic boundary conditions in $y$-direction and open boundary conditions in $x$-direction. To integrate a turbulent regime in the wind-turbine simulations with open streamwise boundary conditions, a synchronized coupling between the EBL/SBL regime and the wind-turbine simulation is applied. Here, the initial fields of

$\psi = (u, v, w, \Theta^{'})$ (Fig. 3(a) and (b)) at 1800 UTC/0000 UTC are applied as initialization of the corresponding wind-turbine simulation. Their horizontal averages are the background fields $\psi_e(z) = (u_e(z), v_e(z), w_e(z), \Theta_e(z))$. At each timestep, two dimensional $y$-$z$ slices represent the upstream values of $\psi$ at the left-most edge of the domain. This technique is similar to Kataoka and Mizuno (2002), Naughton et al. (2011), Witha et al. (2014), and Dörenkämper et al. (2015) and has successfully been applied in Englberger and Dörnbrack (2018), where also a more detailed description is provided.

Further, the wind-turbine simulations are performed on $512 \times 512 \times 64$ grid points with a horizontal resolution of 5 m and a vertical resolution of 5 m in the lowest 200 m and 10 m above. The rotor with $R = 50$ m and $z_h = 100$ m is located at 300 m in x-direction and centered in y-direction. The wind-turbine simulations are performed with a rotation frequency of $\Omega = 0.117$ s$^{-1}$ for 10 min and after restart continued for 50 min.

Three different working conditions of the rotor are considered: A common clockwise rotating rotor (CR), a counterclockwise rotating rotor (CCR), and no rotation of the rotor (NR), defined from a view looking downwind on the turbine (Fig. 2). In the simulations of the considered working conditions in the same atmospheric regime, only the prefactors $\beta_v \in \{ 0, -1, 1\}$ (Eq. 1) differ, as explained above.

## 3  Analysis of a rotating system under veering inflow

The aim of this work is to investigate if the rotational direction of a wind turbine impacts the wake. As a first step, we summarize the basic physics underlying the interaction process of a rotating system with a veering inflow.

A rotating system can be described by a Rankine vortex, which combines a solid body rotation describing the rotating system with radius $R$, with a potential vortex for $r > R$. The rotational velocity of the system is expressed by $\omega$. The radial dependence of the tangential velocity is given by:

$$v_{tangential}(r) = \begin{cases} \omega r & r \leq R \\ \omega \frac{R^2}{r} & r > R \end{cases}$$

Consider a rotating system with a rotation area perpendicular to the $x$-direction (west-east). The flow components (subscript f, $_f$) in meridional $v_f$ and vertical $w_f$ direction interact with the rotating system and are modified by the velocity component of the vortex (subscript v, $_v$) $v_v$ and $w_v$. The interaction of the flow and the vortex components modify the flow in the wake $v$ and $w$ by:

$$v_v = \pm \omega r sin(\Theta), \tag{9}$$

$$w_v = \mp \omega r cos(\Theta), \tag{10}$$

where the signs $\pm$ in Eq. 9 and $\mp$ in Eq. 10 define the rotational direction of the flow in the wake.

For an inflow, interacting with the rotating system, the Ekman spiral (Eqs. 7 and 8) is applied as

$$u_f(z) = u_{Ekman}(z), \tag{11}$$

Regarding the meridional wind component, two cases are possible:

$$CaseA: \qquad \frac{\partial v}{\partial z} = 0 \rightarrow v_f(z) = 0 \tag{12}$$

$$CaseB: \qquad \frac{\partial v}{\partial z} \neq 0 \rightarrow v_f(z) = v_{Ekman}(z). \tag{13}$$

The vertical velocity is

$$w_f(z) = 0. \tag{14}$$

Assuming a simple linear superposition of the atmospheric flow with the Rankine vortex, the interaction process modifies the flow in the wake according to:

$$v(z) = v_f(z) + v_v(z),$$

$$w(z) = w_f(z) + w_v(z).$$

In the following, we focus on the meridional flow component $v$ (Eqs. 12 and 13). We do not need to consider the vertical flow component $w$ as there is no difference between case A and B in Eq. 14, resulting in Eq. 10 for both cases. Only the $v$-component differs between case A (Eq. 12) and B (Eq. 13).

$$CaseA: v(z) = v_v(z) = \pm \omega r sin(\Theta) \tag{15}$$

$$CaseB: v(z) = v_f(z) + v_v(z) = u_g \cdot exp(-z\gamma) sin(z\gamma) \pm \omega r sin(\Theta) \tag{16}$$

The rotational direction of the system (defined by $+$ or $-$ in Eq. 9 and $-$ or $+$ in Eq. 10) has a significant impact on $v(z)$, especially in case B (Eq. 16). Depending on the sign of $v_v(z)$, the meridional velocity of the inflow $v_f(z)$ can either be intensified or decelerated or even reversed in the wake. This modification of $v(z)$ peaks directly behind the rotating system, as the strength of the vortex induced by the rotor $v_v(z)$ decreases downwind as the flow conditions in the wake approach the

inflow value $v_f(z)$. The position of complete wake recovery $x_\zeta$ is defined as the downwind position where $v_v(z)$ approaches zero and $v(z) = v_f(z)$. To simplify the complex wake entrainment process, we define a linear decrease of $v_v(z)$ between the rotating system and downwind distance $x_\zeta$ by Eq. 17, for a given downstream distance from the rotating system $x_{down}$:

$$v(z, x_{down}) = \begin{cases} caseA: & \begin{cases} \pm \omega r sin(\Theta)(1 - \frac{x_{down}}{x_\zeta}) & x_{down} < x_\zeta \\ 0 & x_{down} \geq x_\zeta \end{cases} \\ caseB: & \begin{cases} u_g \cdot exp(-z\gamma) sin(z\gamma) \pm \omega r sin(\Theta)(1 - \frac{x_{down}}{x_\zeta}) & x_{down} < x_\zeta \\ u_g \cdot exp(-z\gamma) sin(z\gamma) & x_{down} \geq x_\zeta \end{cases} \end{cases} \tag{17}$$

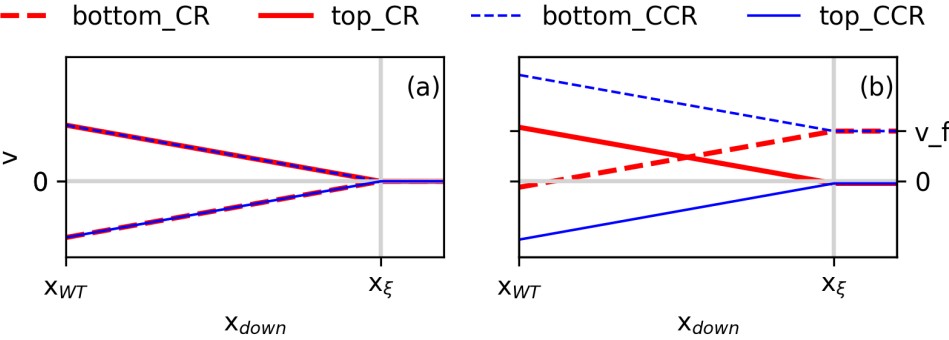

**Figure 4.** Downstream dependence of $v(z, x_{down})$ at $z = z_h - R$ as 'bottom' and $z = z_h + R$ as 'top' considering a center of the rotating system at $z_h$ with a radius $R$. $CR$ and $CCR$ correspond to the clockwise and counterclockwise rotational direction of the rotor.

We can consider the wake behaviour in the lower rotor half separately from the wake behaviour in the upper rotor half of a rotating system, with a rotor center $z_h = 100$ m and a rotor radius $R = 50$ m (Figure 4). In Figure 4(a), we see case A while in Fig. 4(b), we see case B from Eq. 17. The profiles labeled as bottom correspond to $v(z_h - R, x_{down})$ and top to $v(z_h + R, x_{down})$. The inflow in case B is characterised by the Ekman spiral (Eqs. 7, 8), resulting in a directional shear $ds$ over

the lower rotor part of $0.23°$ m$^{-1}$ (Fig. 3(a)). The revolutions per minute of the wake vortex are set to $1/5ds = \omega = 0.046$ s$^{-1}$ for both cases.

For both case A and case B, the wake behaviour at top and bottom of the rotor depends on the sign applied in Eq. 17. If '+' is applied in the lower rotor part of case A (Fig. 4(a)), $v_v(z_h - R, x_{down})$ flows from north to south ($v_v(z_h - R, x_{down}) = +\omega r \sin(270°) = -\omega r < 0$), corresponding to a counterclockwise wake rotation arising from a clockwise rotor rotation (CR).

The interaction of the inflow with the vortex leads to $v(z_h - R, x_{down}) < 0$ m s$^{-1}$ in the near wake and a linear increase up to $v(z_h - R, x_\xi) = v_f(z_h - R) = 0$ m s$^{-1}$ at $x_{down} = x_\xi$. If '-' is applied in the lower rotor part of case A (Fig. 4(a)), $v_v(z_h - R, x_{down})$ flows from south to north ($v_v(z_h - R, x_{down}) = -\omega r \sin(270°) = \omega r > 0$). This situation corresponds to a clockwise wake rotation and a counterclockwise rotor rotation (CCR). In case of CCR, the interaction of the inflow with the vortex leads to $v(z_h - R, x_{down}) > 0$ m s$^{-1}$ in the near wake and a linear decrease up to $v(z_h - R, x_\xi) = v_f(z_h - R) = 0$ m s$^{-1}$ at $x_{down} = x_\xi$.

In the upper rotor half of case A, the situation for $v(z_h + R, x_{down})$ is the same as in the lower rotor half, only with the opposite sign of $v(z_h - R, x_{down})$ applied in Eq. 17 (Fig. 4(a)).

Veer can be constrained to the lower half of the rotor, as seen in case B (Fig. 4(b)). Then it is characterized by a zonal wind from west to east ($u_f(z_h - R) > 0$) with a meridional component from south to north ($v_f(z_h - R) > 0$) in the lower rotor half and a meridional wind component of zero ($v_f(z_h + R) = 0$ m s$^{-1}$) in the upper rotor half. In the following, the situation

for the lower rotor half is discussed, as the upper rotor half of case B corresponds to the behaviour of the upper rotor half of case A. In case of CCR, $v_v(z_h - R, x_{down}) > 0$, and therefore also flows from south to north. The vortex $v_v(z_h - R, x_{WT} + \epsilon)$ intensifies the inflow $v_f(z_h - R)$, resulting in $v(z_h - R, x_{WT} + \epsilon) = v_f(z_h - R) + v_v(z_h - R, x_{down})$ directly behind the rotor, with

$\epsilon \ll R$. Approaching downwind, the vortex impact $v_v(z_h - R, x_{down})$ and therefore also the flow in the wake $v(z_h - R, x_{down})$ decreases approaching $v_f(z_h - R)$ at $x_\xi$.

In case of CR, $v_v(z_h - R, x_{down}) < 0$ m s$^{-1}$, and therefore flows from north to south, weakening the inflow $v_f(z_h - R)$. Therefore, in comparison to $v_f(z_h - R)$, $v(z_h - R, x_{down})$ decreases in the case of $|v_v(z_h - R, x_{down})| < |v_f(z_h - R)|$ or even reverses the sign if $|v_v(z_h - R, x_{down})| > |v_f(z_h - R)|$, as it is the case in Fig. 4(b). Approaching downwind, the vortex impact $v_v(z_h - R, x_{down})$ decreases and therefore the flow in the wake $v(z_h - R, x_{down})$ increases up to $x_\xi$ where $v(z_h - R, x_\xi) = v_f(z_h - R)$ and $x_v(z_h - R, x_\xi) = 0$ m s$^{-1}$.

Summarizing, the analysis predicts a significant difference in the wake flow of a rotating system for a clockwise rotating rotor in comparison to a counterclockwise rotating rotor, but only in the case of a veering inflow (case B).

## 4   A rotating wind-turbine rotor under veering inflow

Of course, this rotating rotor interacting with a veering inflow represents a wind turbine operating in nighttime stably stratified atmospheric conditions. To investigate the impact in a turbulent atmosphere for both cases presented in the analysis, here the non-veering inflow of an EBL situation and the veering inflow of an SBL situation are considered. For numerical details of the EBL and SBL precursor simulations and the corresponding wind-turbine simulations, we refer to Sect. 2.

### 4.1   Impact on spanwise and vertical velocity

As suggested by the analysis of Sect. 3, the rotational direction impact is limited to the change of the sign in Eq. 17 when the inflow lacks veer, as in the EBL. This behaviour is investigated by assessing the wake characteristics of an EBL wind-turbine simulation, which are presented in the $y$-$z$-plane $x = 3$ D or 5 D in Fig. 5. The top row (Fig. 5(a) - (d)) represents the vectors $(v, w)$. As expected, the rotational direction of the flow in the wake inside the rotor region is dictated by the rotational direction of the rotor itself and is opposite to the rotor rotation. Approaching downwind from $x = 3$ D to 5 D, the rotational direction imprint of the wind-turbine wake decreases.

The interactions between the wake rotation and the inflow are embodied in the crossstream and vertical velocities. The evolution $v$ and $w$ dictated, is represented in the second row ((e) - (h)) for $v$ and in the third row ((i) - (l)) for $w$ in Fig. 5. As predicted by the analysis of Sect. 3 (Eq. 15), the signs of $v$ are opposite in the upper and the lower rotor half for CR and CCR, respectively (compare Fig. 5(e) to (g) and (f) to (h)). The same is valid for the signs of $w$ (Eq. 10) (compare Fig. 5(i) to (k) and (j) to (l)). This difference between CR and CCR is pronounced at a downwind distance of $x = 3$ D. At $x = 5$ D, the difference is smaller as $v$ and $w$ approach the inflow conditions of $\approx 0$ m s$^{-1}$ (Fig. 3(a)).

In contrast to the no veer case, the effect of the veering inflow on the wind-turbine wake will not be limited to the change of the sign, also the wake characteristics will be different in CR and CCR, as suggested by the analysis of Sect. 3. Figure 6 represents the wake characteristics for a veering inflow in the SBL wind-turbine simulations. Here, the top row ((a) - (d)) represents the $(v, w)$ vectors in $y$-$z$-plane as difference between the quantities at $x = 3$ D (or 5 D) and upstream at $x = -2$ D to emphasize the effect of the rotating actuator. At a downwind distance of $x = 3$ D, the rotational direction of the rotor determines

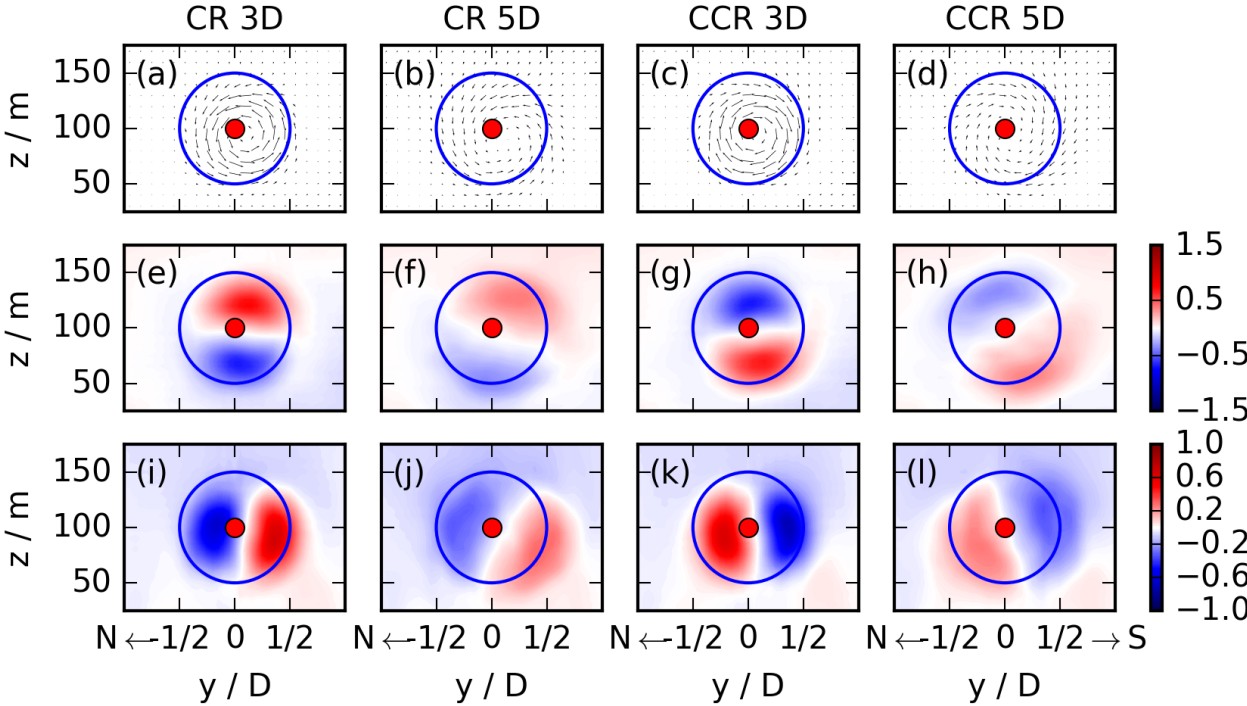

**Figure 5.** $y$-$z$-cross sections of the EBL wind-turbine simulations for CR and CCR at $x = 3\,\mathrm{D}$ and $5\,\mathrm{D}$. The first row ((a)-(d)) presents the $(v, w)$ vectors in the $y$-$z$-plane, the second row ((e)-(h)) the meridional wake velocity $v$, and the third row ((i)-(j)) the vertical wake velocity $w$. The blue contour represents the circumference of the actuator. The streamwise flow is from the west and these cross sections correspond to a view towards the east downwind through the turbine with north on the left and south on the right.

the wake rotation in both cases. Differences emerge by 5 D downwind. The wake rotation in CCR is the same as at $x = 3\,\mathrm{D}$, but only the magnitude of the $(v, w)$ vectors is smaller. In CR, however, the magnitude of $(v, w)$ approaches zero and the rotational direction of the rotor no longer determines the wake rotation. Further, the applied vortex leads to an inflow into the lower northern rotor part, resulting in rising motion in the southern rotor part in CR (Fig. 6(a)), whereas is ascends in the northern rotor part in CCR (Fig. 6(c)).

The second row (Fig. 6(e)-(h)) corresponds to the first row of Fig. 5, representing the vectors $(v, w)$ at $x = 3\,\mathrm{D}$ and $5\,\mathrm{D}$. The magnitude is weaker in CR (Fig. 6(e)) in the upper and lower rotor part in comparison to CCR (Fig. 6(g)), resulting in larger and more organised $(v, w)$ wind components in CCR. At a downwind distance of $x = 5\,\mathrm{D}$, in CCR (Fig. 6(h)), the rotational direction of the wake flow is the same as at $x = 3\,\mathrm{D}$, only the strength is weaker. In case of CR (Fig. 6(f)), however, the flow direction in the upper rotor half has changed in comparison to $x = 3\,\mathrm{D}$ (Fig. 6(e)). Now it flows from north to south, resulting in a wake which is clearly dominated by the inflow.

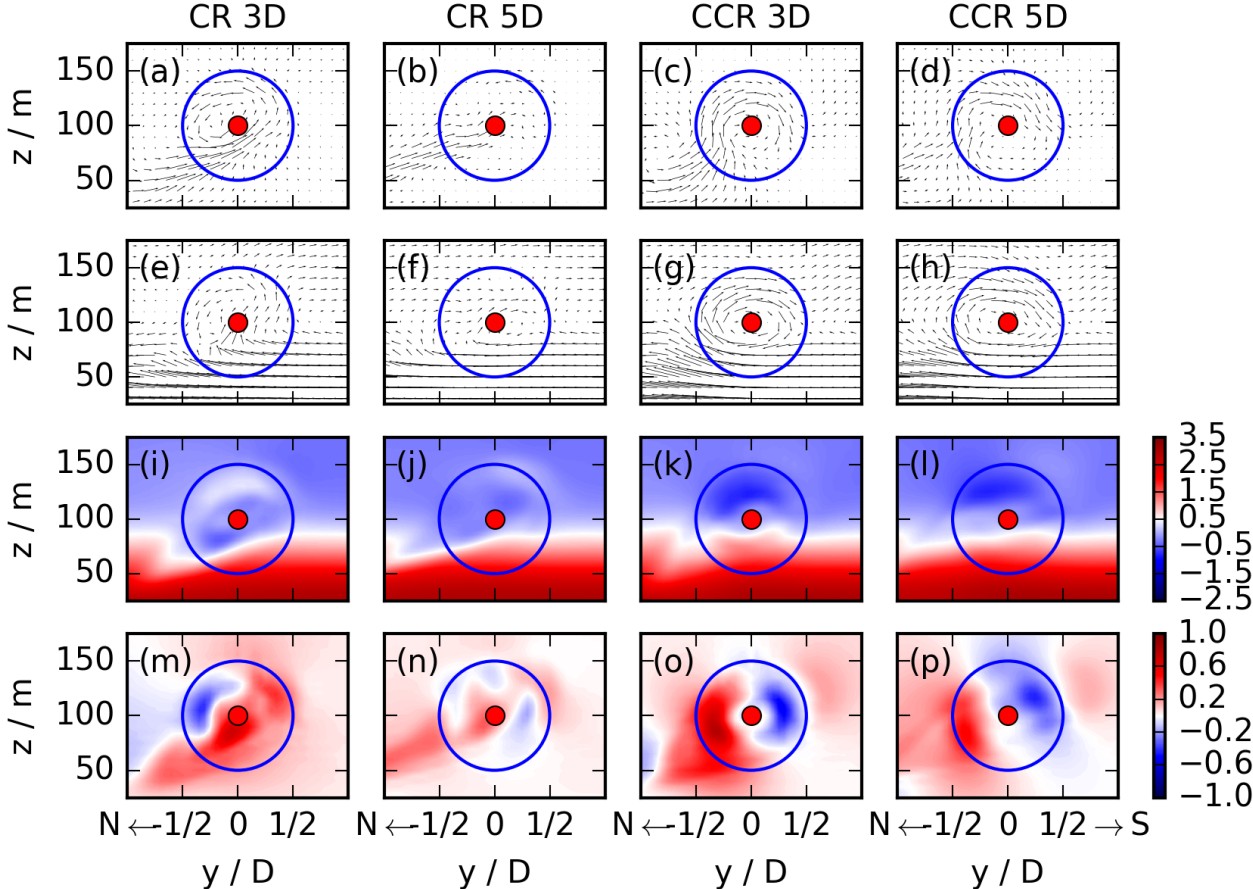

**Figure 6.** $y$-$z$-cross sections of the SBL wind-turbine simulations for CR and CCR at $x = 3\,\mathrm{D}$ and $5\,\mathrm{D}$. The first row ((a)-(d)) presents the $(v, w)$ vectors as difference between the vector at $x = 3\,\mathrm{D}$ (or $5\,\mathrm{D}$) and the inflow region, the second row ((e)-(h)) represents the velocity vector in the $y$-$z$-plane, the third row ((i)-(l)) the meridional wake velocity $v$, and the fourth row ((m)-(p)) the vertical wake velocity $w$. The blue contour represents the circumference of the actuator. The streamwise flow is from the west and these cross sections correspond to a view towards the east downwind through the turbine with north on the left and south on the right.

To distinguish both contributing parameters of the $(v, w)$ vectors, the $v$-cross sections are presented in the third row ((i)-(l)) of Fig. 6. The positive and negative perturbations in $v$ are the opposite in CR (Fig. 6(i)) and CCR (Fig. 6(k)) in the corresponding rotor sector at $x = 3\,\mathrm{D}$, with larger $|v|$ values in the upper and lower rotor sector in case of CCR. This pattern is weakening downwind and persists at $x = 5\,\mathrm{D}$ (Fig. 6(j), (l)).

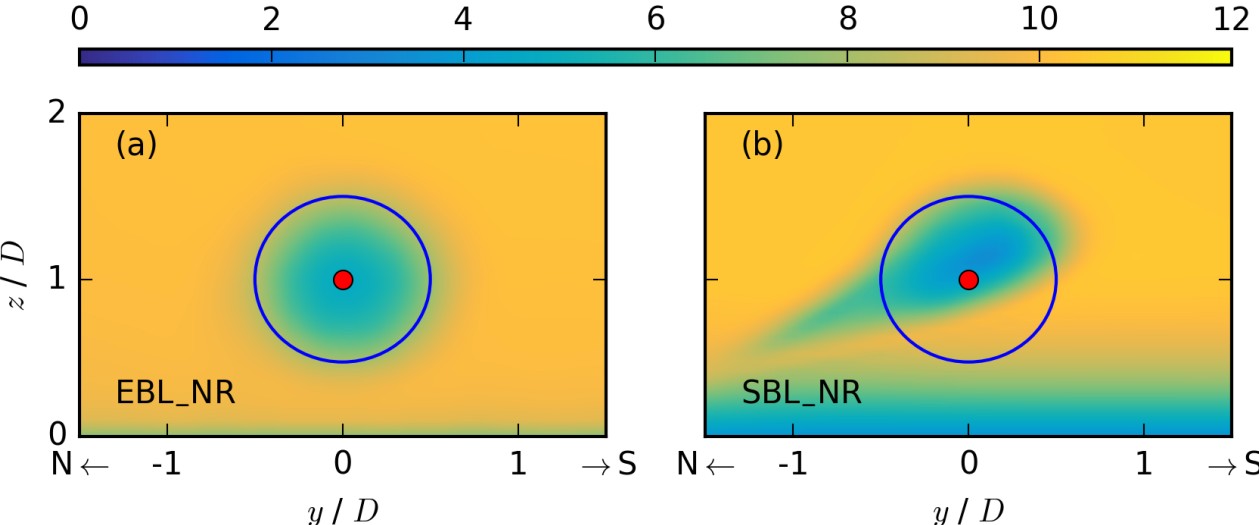

**Figure 7.** Coloured contours of the time averaged zonal velocity $\overline{u_{i_*,j,k}}$ in m s$^{-1}$ at a downward position of $x = 5\,\mathrm{D}$ with indes $i_*$ behind the rotor of the non-rotating EBL wind-turbine simulation without wind veer in (a) and the non-rotating SBL wind-turbine simulation with veering inflow in (b). The blue contour represents the circumference of the actuator.

Considering the $w$-cross section in the fourth row ((m)-(p)) of Fig. 6, the upward and downward orientation of $w$ differs at $x = 3\,\mathrm{D}$ (Fig. 6(m), (o)) in the northern and the southern sector and also for CR (Fig. 6(m)) and CCR (Fig. 6(o)) comparing the same sectors. In comparison to the EBL wind-turbine simulations (Fig. 5(i)-(l)), the veering inflow has an additional effect on the vertical velocity in the wake at $x = 3\,\mathrm{D}$ (Fig. 6(m) and (o)). The veering inflow leads to an intensified wake entrainment at the lower northern rotor part, which is independent of the rotational direction (see inflow pattern of the $(v, w)$ vectors in the lower northern rotor part in Fig. 6(a) and (c)). At this location the flow is directed upward. In case of CCR, it overlaps with the updraft in the northern sector, resulting in a rising motion of $w$ in the northern part at $x = 3\,\mathrm{D}$ (Fig. 6(o)). In case of CR, a new updraft region manifests, weakening the downward region in the northern rotor part. This interaction results in an updraft in the southern rotor part at $x = 3\,\mathrm{D}$ (Fig. 6(m)). At $x = 5\,\mathrm{D}$ (Fig. 6(n), (p)), the flow pattern is the same as at $x = 3\,\mathrm{D}$, only the strength is weaker.

## 4.2   Impact on streamwise velocity and total turbulence intensity

The rotational direction of the rotor modifies the spanwise and vertical velocity components in the wake under veering inflow. Here, we investigate their effect on the streamwise velocity and the total turbulence intensity.

The inflow profiles from Fig. 3(a) predict a different wake behaviour under veering inflow in comparison to no wind veer, regardless of the rotational direction. Therefore, different wake characteristics should prevail in both non-rotating simulations NR. Figure 7 represents $y$-$z$-cross sections of the streamwise velocity at $x = 5\,\mathrm{D}$ for the non-rotating wind-turbine simulations

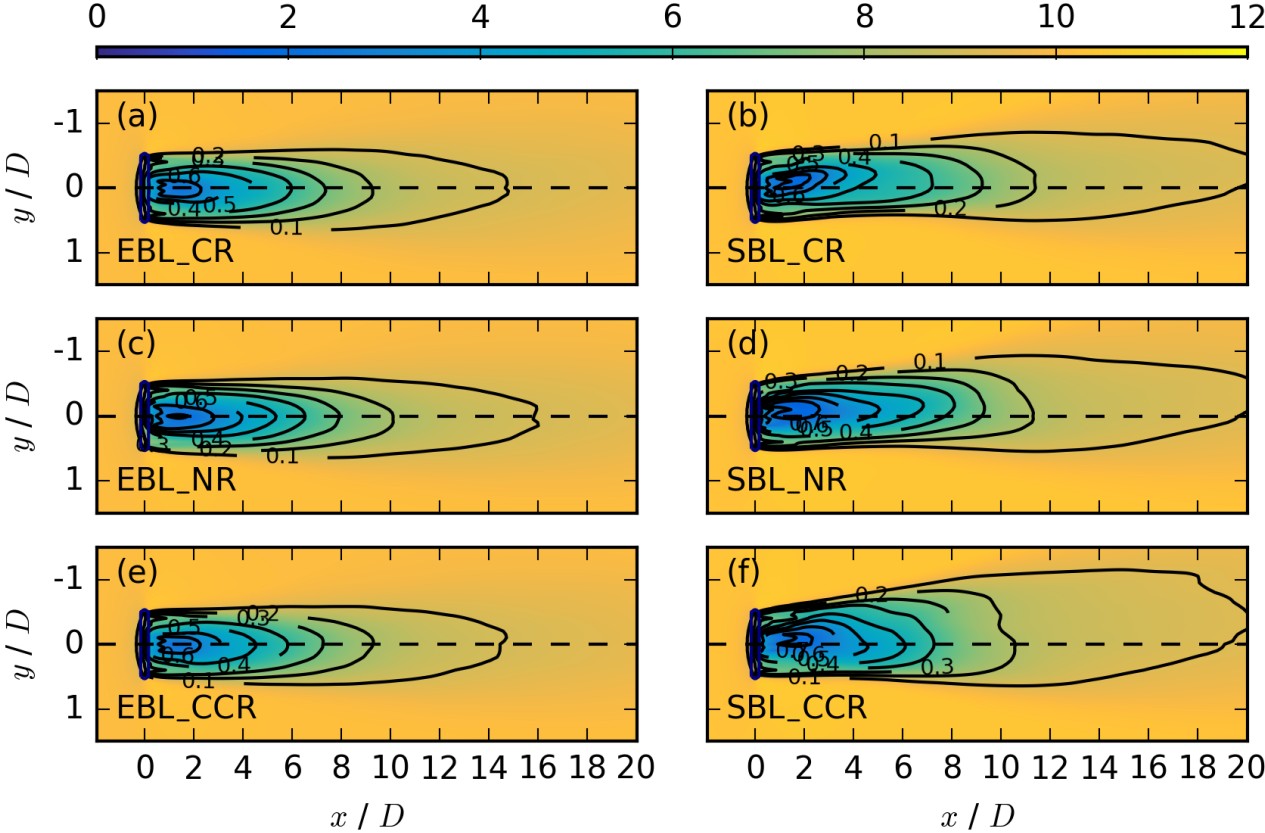

**Figure 8.** Coloured contours of the streamwise velocity $\overline{u_{i,j,k_h}}$ in m s$^{-1}$ at hub height $k_h$, averaged over the last 50 min, for EBL_CR in (a), SBL_CR in (b), EBL_NR in (c), SBL_NR in (d), EBL_CCR in (e), and SBL_CR in (f). The black contours represent the velocity deficit $VD_{i,j,k_h}$ at the same vertical location.

with a non-veered inflow in the evening in (a) and a veering inflow in a stable regime in (b). In case of no veer, the wake at $x = 5$ D retains the shape of the rotor (Fig. 7(a)). In the case of a veering inflow, the wake in the lower rotor half shifts to the north (Fig. 7(b)). The striking difference in the lower rotor part corresponds to the inflow profiles of Fig. 3(a), where a veering inflow is characterized by a southern component for $z < 100$ m in the SBL, whereas the meridional inflow velocity is $\approx 0$ m s$^{-1}$

5   in the EBL in all rotor heights. The skewed wake is only due to the veering inflow, as it also occurs without a rotating rotor. This simulated structure resembles those of the simulations of Abkar and Porté-Agel (2016), Vollmer et al. (2017), Bromm et al. (2017), Churchfield and Sirnivas (2018) and Englberger and Dörnbrack (2018).

The evolution of the wake at specific altitudes also show the effect of veer. The streamwise velocity appears at hub height of $z_h = 100$ m in Fig. 8, in the upper rotor part at $z = 125$ m in Fig. 9 and in the lower rotor part at $z = 75$ m in Fig. 10 for the

10   EBL (no veer) wind-turbine simulations in the left column ((a), (c), (e)) and the SBL (veered) wind-turbine simulations in the

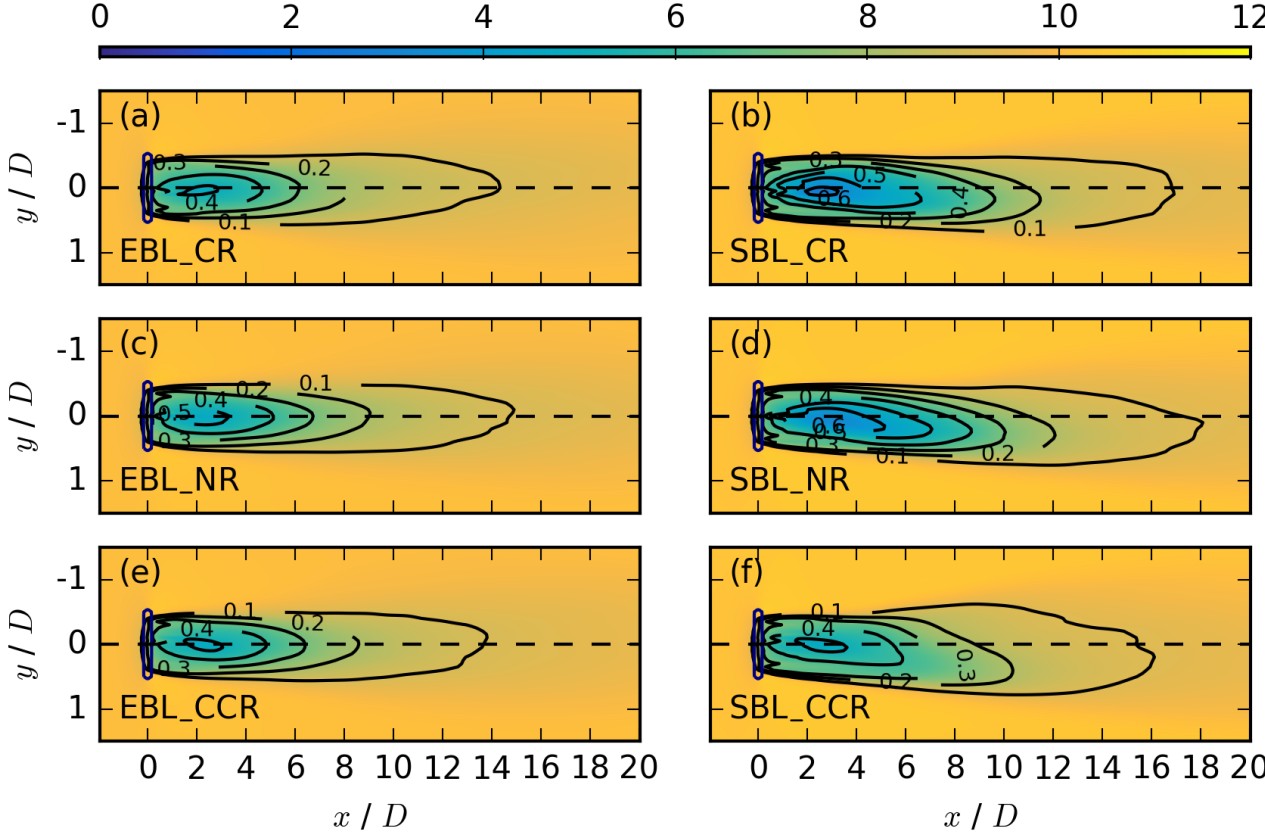

**Figure 9.** Coloured contours of the streamwise velocity $\overline{u_{i,j,k_*}}$ in m s$^{-1}$ at $z_h = 125$ m with index $k_*$, averaged over the last 50 min, for the same cases as in Fig. 8. The black contours represent the velocity deficit $VD_{i,j,k_*}$ at the same vertical location.

right column ((b), (d), (f)) for a clockwise rotating rotor CR in the first row, no rotation of the disc NR in the second row, and a counterclockwise rotating rotor CCR in the third row.

According to the analysis of Sect. 3, disregarding the sign in Eq. 17, no significant wake differences are expected in the non-veering simulations of the EBL for CR and CCR. This is in agreement with the situation at hub height (Fig. 8(a) vs. (e)). Above

5  and below hub height, differences emerge in the near wake (Figs. 9(a), (e), 10(a), (e)), resulting from the opposite sign in CR and CCR in the upper and lower rotor part (Eq. 17). The near wake deflects towards the north in the upper rotor half (Fig. 9(a)) and towards the south in the lower rotor half (Fig. 10(a)) in CR, whereas it deflects towards the north in the lower rotor part (Fig. 10(e)) and towards the south in the upper rotor part (Fig. 9(e)) in CCR. In contrast, the non-rotating EBL simulation NR does not show any near wake deflection at any height (Figs. 8(c) - 10(c)). The effect in the rotating actuator simulations is

10  therefore caused by the rotation of the rotor, and can be explained by a transport of higher momentum air counterclockwise (clockwise) in CR (CCR), with the opposite situation prevailing at 75 m.

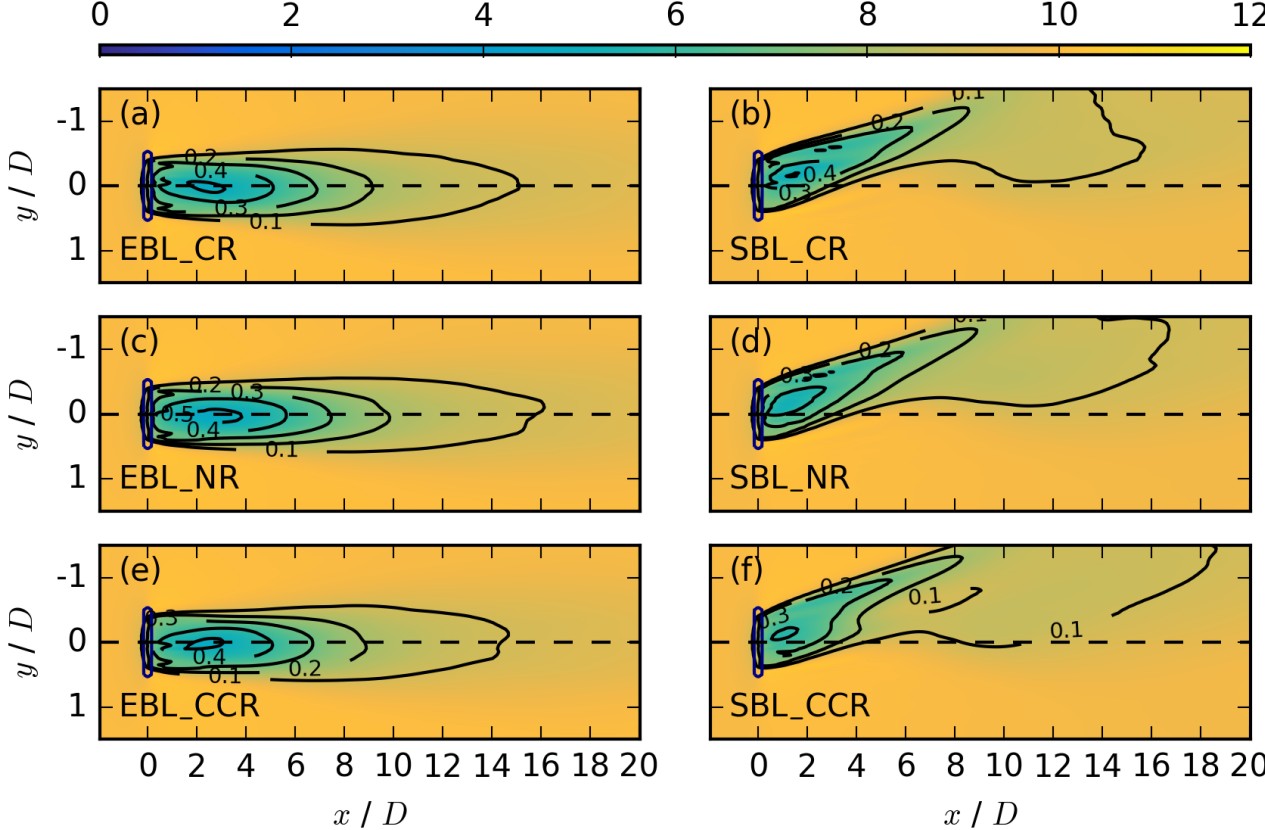

**Figure 10.** Coloured contours of the streamwise velocity $\overline{u_{i,j,k_*}}$ in m s$^{-1}$ at z = 75 m with index $k_*$, averaged over the last 50 min, for the same cases as in Fig. 8. The black contours represent the velocity deficit $VD_{i,j,k_*}$ at the same vertical location.

Wake elongation also exhibits the impact of the rotor's rotation. Comparing NR to CCR and CR of the EBL regime, there is further a difference in the wake elongation, with a larger velocity deficit in NR at hub height (Fig. 8(c)) at the same downwind position in comparison to CCR and CR (Fig. 8(e) and (a)). The faster recovery is related to enhanced entrainment in the CR and CCR simulations as the rotation itself acts as source of turbulence.

5     In case of a veering inflow, the analysis of Sect. 3 predicts significant wake differences in the CR and CCR wind-turbine simulations of the SBL regime, which are not limited to the change of the sign in CR and CCR. The wake structures at hub height, presented in Fig. 8(b) and (d) for the CR and the NR simulations are comparable regarding the elongation, the width, and the deflection angle, respectively. The wake structure in CCR (Fig. 8(f)) differs. Considering the black velocity deficit contours (Eq. 4), the wake recovers more rapidly and, especially in the far wake, it is broader with a slightly larger wake

10     deflection angle towards the dashed center line.

The same situation occurs in the upper rotor part at 125 m (Fig. 9). Under veering wind conditions, the wake characteristics are also comparable in CR (Fig. 9(b)) and NR (Fig. 9(d)), with a slightly less rapid wake recovery in NR due to no additional turbulence generated by disc rotation. Comparing CR (and NR) to CCR (Fig. 9(f)), the wake recovers much more rapidly in CCR, with a smaller velocity deficit in the near wake with $VD_{max} \approx 0.4$ in comparison to $VD_{max} \approx 0.6$ in CR and NR.

A comparison of NR EBL (Fig. 9(c)) to NR SBL (Fig. 9(d)) shows a wake deflection angle in the NR SBL simulation, resulting from a meridional wind component $\leq 0$ m s$^{-1}$ in the upper rotor part (Fig. 3(a)). Comparing the rotating SBL cases (Fig. 9(b), (f)) to NR (Fig. 9(d)), the wake deflection angle decreases in CR, whereas it increases in CCR. This deflection results from the counterclockwise transport of higher momentum air in CR and the clockwise transport in CCR. Interestingly, although veer is constrained to the lower half of the rotor disk, these differences in the upper half of the wake still occur.

According to the analysis of Sect. 3, the effect of the rotational direction on $v$ and $w$ is predicted to be much larger in the lower rotor part, and this is also expected for $u$. Figure 10(b), (d), and (f) represent the situation in the lower rotor part at 75 m. CR (Fig. 10(b)) and NR (Fig. 10(d)) again show similar wakes, however, comparing CR (and NR) to CCR (Fig. 10(f)), the velocity deficit maximum, the elongation, the width, and the deflection angle of the wakes differ. Compared to the difference in the wake deflection angle in the near wake for CR and CCR in the EBL wind-turbine simulations (Fig. 10(a) and (e)), the wake deflection angle in the SBL wind-turbine simulations (Fig. 10(b) and (f)) is also influenced by the inflow wind direction angle. In case of CR, the wake deflection angle decreases (Fig. 10(b)), whereas in CCR, it increases (Fig. 10(f)), similar as in the upper rotor part (Fig. 9(b) and (f)).

For a more qualitative investigation of the impact of the rotational direction on the wake under veering inflow, Fig. 11 shows vertical and spanwise profiles through the rotor center of the streamwise velocity in the first two rows ((a)‑(h)) and also of the total turbulence intensity (Eq. 6) in the last two rows ((i)‑(p)) at $x = 3$ D, $5$ D, $7$ D, and $10$ D for CR and CCR rotating turbines in the SBL (solid red (CR) and blue (CCR) lines) and the EBL (dashed red (CR) and blue (CCR) lines).

Without veering inflow, the difference in $u$ and $TI$ between CR and CCR is trivial. Further, the profiles are almost symmetric to the rotor center lines of $z = z_h$ and $y = 0$ D. However, under veering wind conditions in the SBL, significant differences emerge. Considering the upper rotor part up to $z = 150$ m in the vertical profiles of $u$ (Fig. 11(a)‑(d)) and $TI$ (Fig. 11(i)‑(l)), the downwind velocity in case of CCR is larger in comparison to CR, whereas the total turbulence intensity is larger up to $x = 7$ D and smaller at $x = 10$ D. In the top-tip region of the rotor and for $z \geq 150$ m, the total turbulence intensity is larger in case of CR in comparison to CCR for $x \geq 5$ D. To investigate the impact in the lower rotor part, a vertical profile through the rotor center is not appropriate due to wake deflection. Therefore, spanwise profiles of $u$ and $TI$ at 75 m are shown in the first ((a)‑(d)) and third ((i)‑(l)) row of Fig. 12, including the deflected wake towards the north at $y < 0$ D. Up to at least $x = 7$ D (Fig. 12(a)‑(c)), the streamwise velocity in the deflected wake is larger in the CCR case in the southern rotor part with a minor difference between CR and CCR and slightly larger values in CR in the northern rotor part, whereas differences between the profiles have eroded by $x = 10$ D (Fig. 12(d)). The lateral position $y_*$ of the wake center at $x = 3$ D, $5$ D, $7$ D and $10$ D at $z = 75$ m is shown in Fig. 10(b) for CR and in Fig. 10(f) for CCR. In case of CCR at $x = 3$ D, $y_* \approx$ -1/2 D and at $x = 7$ D, $y_* \approx$ -1 D, with slightly smaller values of $y_*$ in case of CR, resulting from the smaller wake deflection angle in CR in comparison to CCR (compare Fig. 10(b) to (f)). Therefore, comparing $u$ at the corresponding wake center positions in

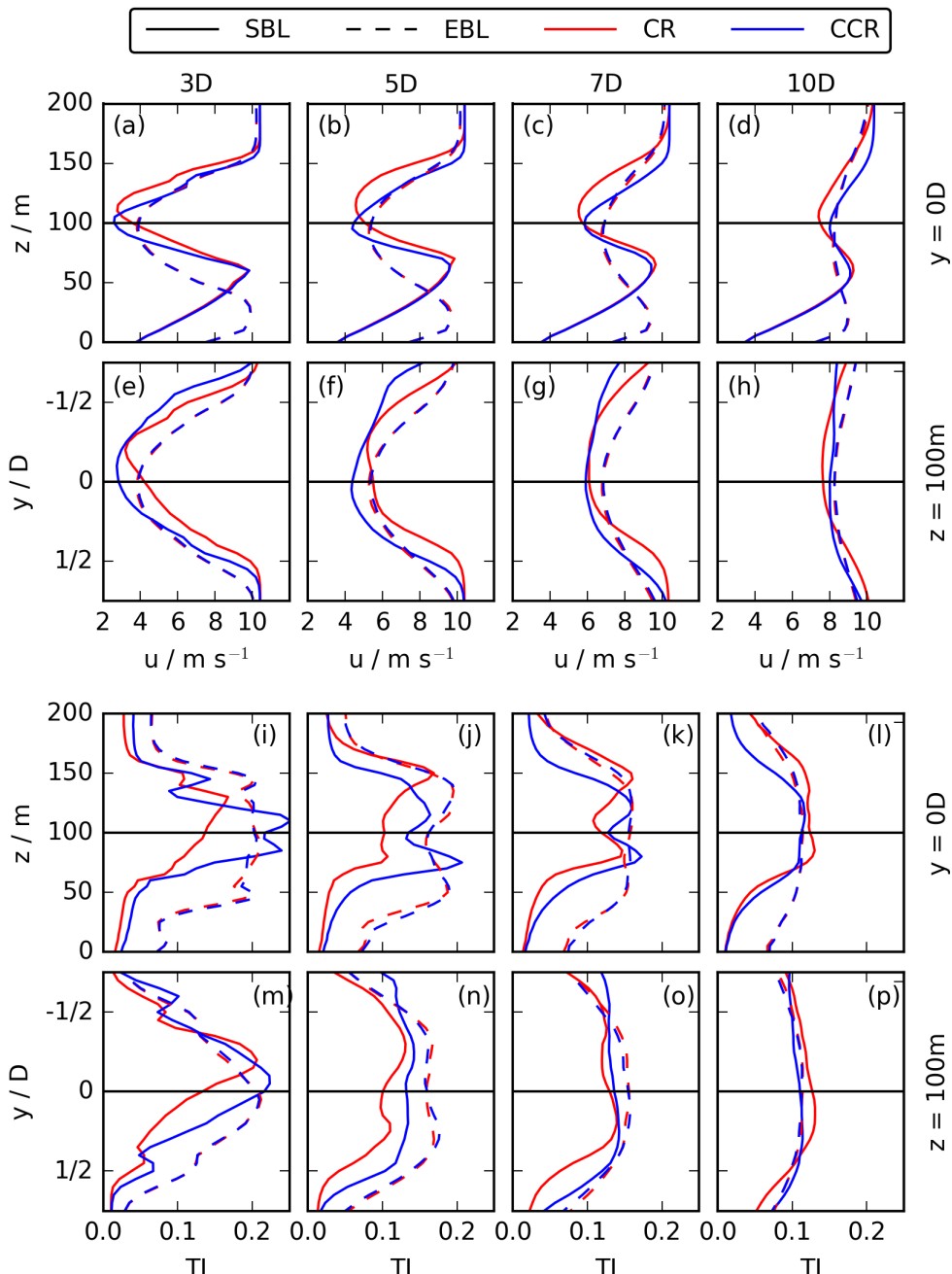

**Figure 11.** Vertical and spanwise profiles of u (first two rows) and $TI$ (last two rows) through the center of the rotor at $y = 0\,\mathrm{D}$ and $z = 100\,\mathrm{m}$ for the SBL and the EBL wind-turbine simulations at $x = 3\,\mathrm{D}$ (first column), $5\,\mathrm{D}$ (second column), $7\,\mathrm{D}$ (third column), and $10\,\mathrm{D}$ (fourth column) downwind and for both rotational directions CR and CCR.

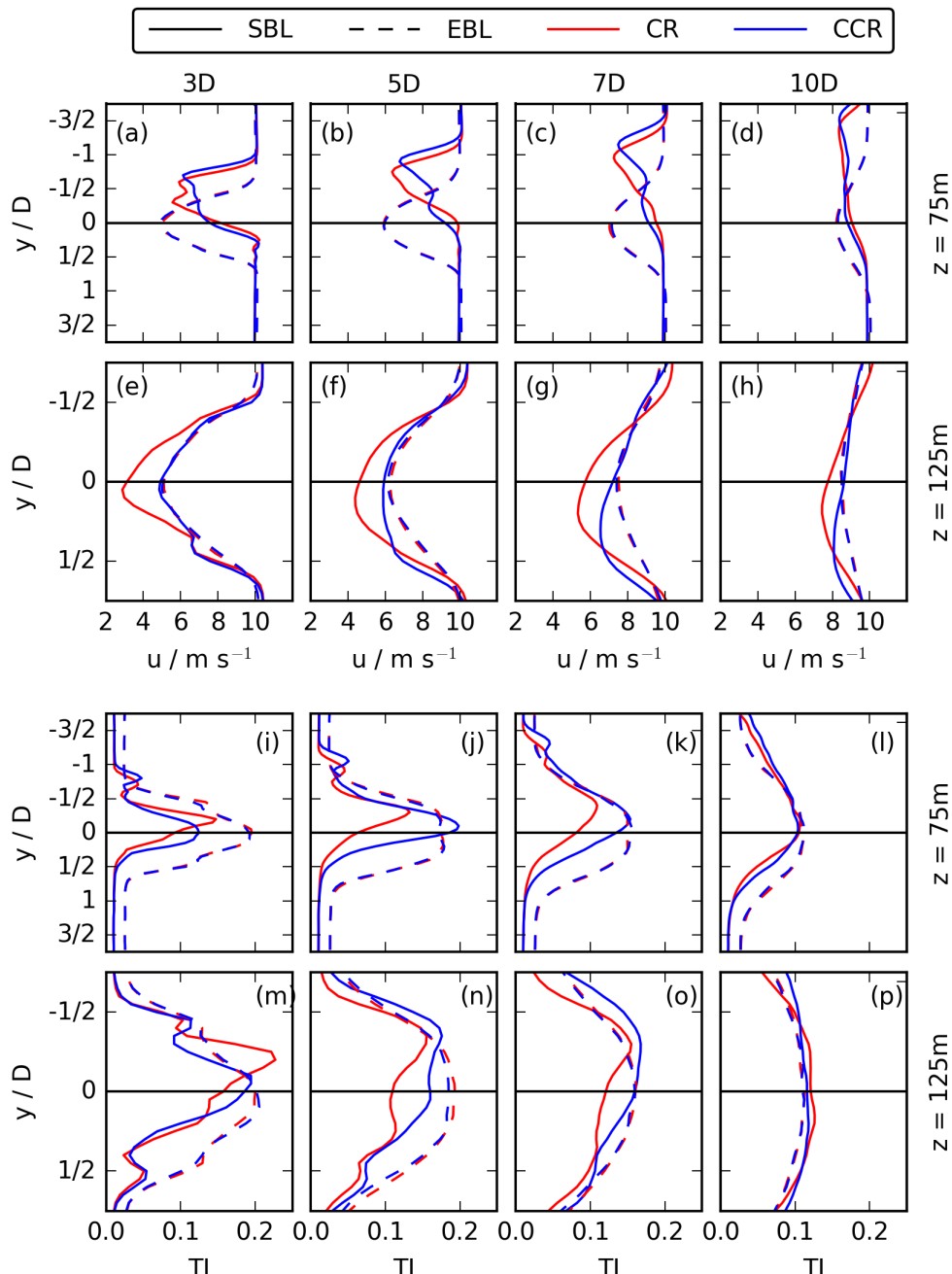

**Figure 12.** Spanwise profiles of u (first two rows) and $TI$ (last two rows) at $z = 75$ m in the first and third row and at $z = 125$ m in the second and last row for the SBL and the EBL wind-turbine simulations at $x = 3$ D (first column), 5 D (second column), 7 D (third column), and 10 D (fourth column) downwind and for both rotational directions CR and CCR.

Fig. 12(a)‑(d), $u_{CCR} > u_{CR}$ for $x \leq 10\,\text{D}$. The same tendency of a higher streamwise velocity in CCR in comparison to CR is valid in the whole wake region at 125 m (Fig. 12(e)‑(h)), where the difference also decreases downwind.

Contrasts in turbulence intensity emerge, due to differences in entrainment. The total turbulence intensity around $y = 0\,\text{D}$ in the lower rotor part (Fig. 12(i)‑(l)) shows the same relation between CR and CCR as in the upper rotor part (Fig. 12(m)‑(p)) and at hub height (Fig. 11(m)‑(p)), with $TI_{CCR} > TI_{CR}$ up to $x = 7\,\text{D}$. However, considering the difference in the deflected wake at $z = 75\,\text{m}$ ((i)‑(l)), $TI_{CCR} < TI_{CR}$ at $x = 3\,\text{D}$ for $y \leq y_* = \text{-}1/2\,\text{D}$ (Fig. 12(i)). Approaching downwind, the difference in $TI$ between CR and CCR decreases for $x \geq 5\,\text{D}$ (Fig. 12(j)‑(l)). The much larger values of CCR at $y = 0\,\text{D}$ in the lower rotor half for $x \leq 7\,\text{D}$ (Fig. 12(i)‑(k)) can be explained with a broader wake width approaching the center line in Fig. 10(f) in comparison to Fig. 10(b) representing CR. Approaching $x = 10\,\text{D}$ (Fig. 12(l)), the profiles for CR and CCR become rather similar, no longer showing a rotational direction impact. Larger $TI$ values in CCR in comparison to CR can be explained by a more rapid wake recovery (Figs. 8 and 10(f) vs. (b)) resulting from a larger entrainment rate in CCR in comparison to CR.

Summarizing, the rotational direction has no significant impact on $u$ and $TI$ in the case of no wind veer, represented by our EBL simulation. However, in case of a veering wind in the SBL, a consistent downwind impact on $u$ and $TI$ emerges at all heights over the rotor, with larger $u$-values and $TI$-values in case of CCR in comparison to CR.

## 5 Comparison of a rotating wind turbine under veering inflow to analysis predictions

The striking difference between the wake characteristics of the no-veer EBL and the veering SBL wind-turbine simulations, and especially the difference between CR and CCR in case of a veering inflow, can be explained by the analysis of Sect. 3 with Eq. 17.

In the upper rotor part, $v_f$ and $v_v$ are the same in the EBL and the SBL wind-turbine simulations. In case of CR, $|\,v_f\,| \approx 0$, interacting with $v_v$ flowing from south to north. As $|\,v_v\,| >> |\,v_f\,|$, it results in a $v$-component flowing from south to north for the EBL in Fig. 5(e), (f) and the SBL in Fig. 6(i), (j). In case of CCR, $v_v$ flows from north to south in the upper rotor part, resulting in a $v$-component also flowing from north to south for the EBL in Fig. 5(g), (h) and in the SBL in Fig. 6(k), (l).

In the lower rotor part, $v_f$ differs in the EBL and the SBL regime (Fig. 3(a)). In case of no wind veer in the EBL, $v_f$ is the same over the whole rotor. Therefore, only the sign of $v_v$ contributing to Eq. 17 changes in the lower rotor half, in comparison to the upper rotor half (Fig. 5(e), (f)). Another effect of $\frac{\partial v}{\partial z} = 0$ are the similar wake characterstics in panels (a) and (e) of Figs. 8‑10. The only minor difference is the wake deflection angle in Figs. 9(a) vs. (e) and 10(a) vs. (e). The non-veering inflow also results in the symmetric structure to hub height and to $y = 0\,\text{D}$ of $u$ and $TI$ in Figs.11 and 12.

In case of veering inflow, $v_f > 0\,\text{m s}^{-1}$ in the lower rotor part and therefore flows from south to north. This pattern leads to a decrease of $v$ in CR due to $v_v < 0\,\text{m s}^{-1}$ (component from north to south) in Fig. 6(i), (j) and an increase in CCR due to $v_v > 0\,\text{m s}^{-1}$ (component from south to north) in Fig. 6(k), (l). Further, the larger velocity deficit values in the horizontal slides in Figs. 8‑10 for (b) vs. (f) and the smaller $u$-vales in the vertical and spanwise profiles up to at least $x = 7\,\text{D}$ in Figs. 11(a)-(d) and 12(a)-(h) in CR in comparison to CCR can be related to the reduction of $v$ in the near wake in CR. Approaching

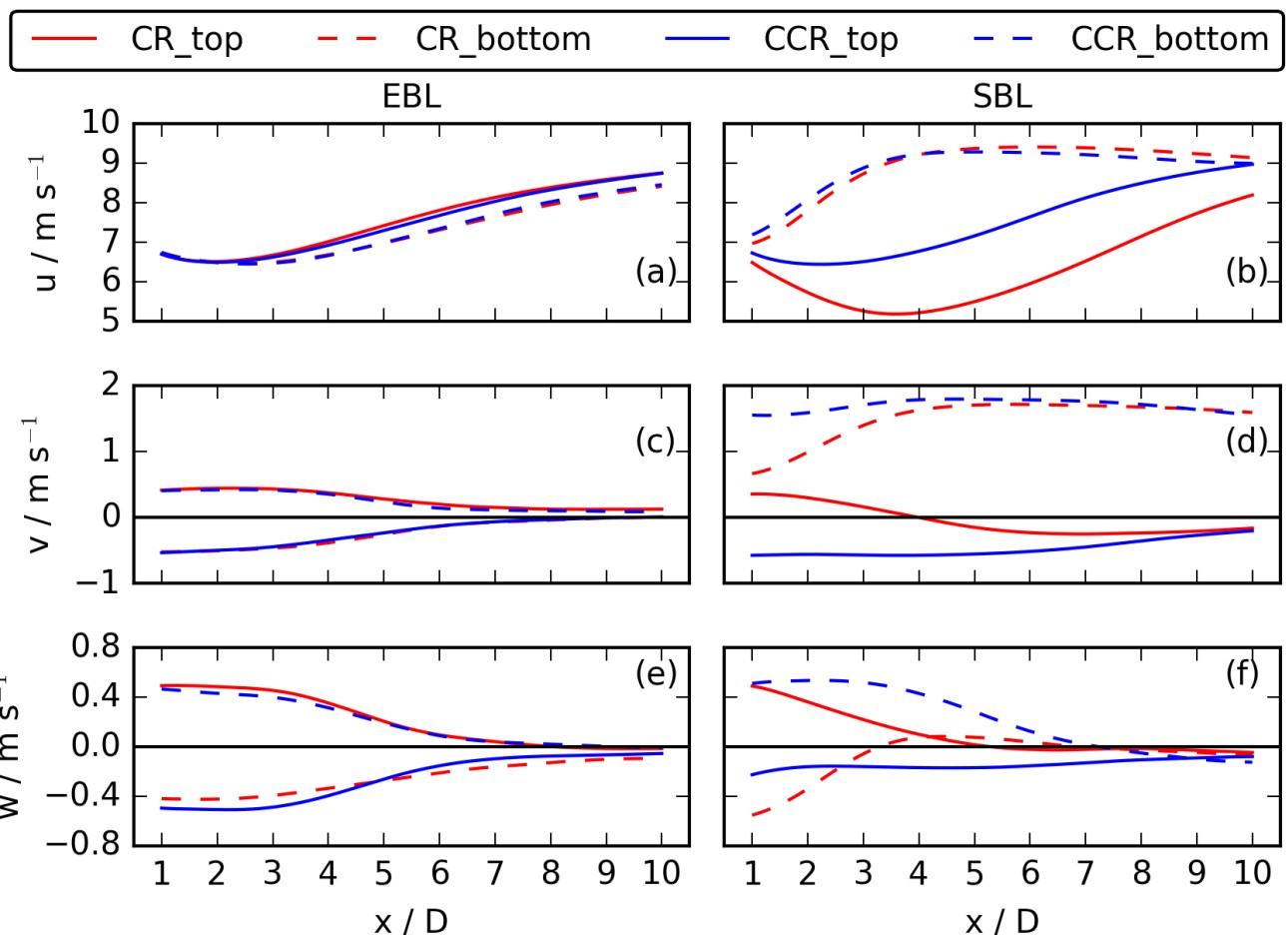

**Figure 13.** Sector and time averages of $u$, $v$, and $w$ for the EBL wind-turbine simulations (first column) and the SBL wind-turbine simulations (second column). $u$ and $v$ show the top and bottom $90°$-sectors, $w$ the southern (solid line) and northern (dashed line) sectors.

downwind, the wake flow approaches the inflow conditions $u_f$, $v_f$, and $w_f$ at $x_{down} = x_\xi > 20\,\text{D}$ in Figs. 8 - 10 for (b) and (f), as the contribution of $v_v$ decreases according to analysis.

The behaviour of the total turbulence intensities (Figs. 11 and 12(i) - (p)) can also be explained by the analysis of Sect. 3. Smaller turbulence intensity values for CR in comparison to CCR can be related to a larger entrainment rate due to $|\,v\,| > |\,v_f\,|$ in CCR, whereas, $|\,v\,| < |\,v_f\,|$ in CR. Moving downwind to $x = 10\,\text{D}$ (Figs. 11 and 12(l), (p)), the difference between $v$ and $v_f$ decreases, resulting in similar values of $TI$ for CR and CCR.

For a direct comparison of the stratified wind-turbine simulations of Sect. 4 with the analysis from Sect. 5, Fig. 13 compares averaged values for the sectors, as sketched in Fig. 2, of all simulated velocity components $u$, $v$, and $w$ up to $x = 10\,\text{D}$ downwind for the no-veer EBL wind-turbine simulations in the left column and the veering SBL wind-turbine simulations in

the right column. We start with a comparison of the non-veered case. The $v$-component of the EBL situation in Fig. 13(c) with $v_f \approx 0$ m s$^{-1}$ is comparable to the non-veering case A and Fig. 4(a). In the near wake, the vortex impact leads to a maximum of $v$. Approaching downwind, the impact of the rotor decreases, approaching to $v_f \approx 0$ m s$^{-1}$ at $x_\xi \approx 7$ D. The only difference to the analysis in Fig. 4(a) is the linear behaviour of $v$ in the near wake up to $x = 3$ D, whereas the approximately linear decrease up to $x = 7$ D is comparable. This near-wake difference results from the complex entrainment process in a turbulent atmospheric boundary layer, which is not considered in the analysis chapter. The behaviour of $w$ of the EBL wind-turbine simulations in Fig. 13(e) shows the same characteristics as $v$ in Fig. 13(c), only for the southern (solid line) and northern (dashed line) sectors. Therefore, the behaviour of $v$ and $w$ corresponds to the expected results from Eqs. 9 and 10. The $u$-values of the EBL wind-turbine simulations in Fig. 13(a) are constant up to $x = 3$ D, followed by a linear increase which flattens at approximately $x = 7$ D. This downwind evolution is comparable to the $v$ and $w$ behaviour and is caused by the entrainment process of turbulent air into the wake region and the resulting wake recovery. The slightly smaller $u$-values in the bottom sector are related to the small increase of $u_f$ over the rotor (Fig. 3(a)).

Comparing the sector averaged values of the EBL wind-turbine simulations (Fig.13(a), (c), (e)) to the SBL wind-turbine simulations (Fig.13(b), (d), (f)), differences are especially pronounced in the CR case. Continuing with a comparison of the rotational direction impact on wake characteristics under veering inflow, it shows striking differences for CR and CCR (Fig.13(b), (f)). The profiles of the $v$-sectors (Fig. 13(d)) are comparable to case B in the analysis chapter and the corresponding Fig. 4(b). For a counterclockwise rotating rotor, $v$ of the near wake is modified by $v_v$, gradually approaching $v_f$ in the far wake. For a clockwise rotating rotor, the $|v|$-component, however, decreases in the near wake as seen in Fig. 6(i). In the lower rotor part, this decrease results from the superposition in Eq. 17 of the inflow from south to north, whereas the $v_v$-component is directed from north to south. Approaching downwind, $|v|$ increases up to $v = v_f$. The upper rotor part is characterized by $v_f \leq 0$ m s$^{-1}$ (Fig. 3(a)). Therefore, $v$ of the top 90°-sector is only influenced by the positive $v_v$-component from south to north. Approaching downwind, the impact of $v_v$ decreases, and $v$ approaches $v_f \leq 0$. The non-symmetric behaviour of the bottom and top sectors in Fig. 13(d) are related to the skewed wake in the lower rotor part in Fig. 7(b). Due to the wake deflection, $v$ approaches $v_f$ at a much smaller downwind distance in comparison to the non-deflected upper rotor part (Figs. 9(b), 10(b)).

Due to the relatively large direction shear of 0.25° m$^{-1}$ in the lower rotor part and a moderate rotational frequency $\Omega$ of 0.117 s$^{-1}$, the $v$-component in the lower rotor part decreases in CR in Fig. 13(d). However, the sign did not change as predicted by Fig. 4(b). We would expect the change to occur for a fast enough rotation of the blades. In fact, the sign change does occur for a doubling of the rotational frequency of the SBL CR wind-turbine simulation (not shown).

The downwind evolution of $w$ in Fig. 13(f) agrees with Eq. (10). The values of the northern and southern sectors differ, and for a clockwise rotating rotor, $w > 0$ m s$^{-1}$ in the southern sector (red solid line), whereas $w < 0$ m s$^{-1}$ in case of a counterclockwise rotating rotor (blue solid line) (Fig. 6(m)-(p)). The $u$-values in Fig. 13(b) show the rotational direction dependence from Figs. 11(a)-(d) and 12(a)-(h), which is much more pronounced in the top sector of Fig. 13(b). The values of $u$ are larger for CCR at each downwind position of the wake in the top sector. This larger difference between CR and CCR in the top sector (Fig. 13(b)) is somewhat surprising as the veering inflow is limited to the lower rotor part in Fig. 3(a). However, the $u$-sections in Fig. 13(b) consider the 90°-sectors in the wake of the rotor (Fig. 2). Due to the veered inflow in the lower rotor

part, the wake is deflected out of the considered sector region roughly at $x = 3\,\mathrm{D}$ (Fig. 10(b), (f)). At $x > 10\,\mathrm{D}$, the $u$-values of the top sector become larger than the bottom sector ones (not shown), related to the significant increase of $u_f$ with height over the rotor part (Fig. 3(a)).

## 6 Conclusions

To investigate the interaction of a veering wind with the rotational direction of a wind turbine in the wake in a stably stratified atmospheric boundary layer, we carry out a series of six LESs by applying incoming wind conditions from a non-veering EBL and a veering SBL extracted from a diurnal cycle simulation. The flow structures of the wake in the LESs are controlled by the rotational direction of the wind-turbine blades imposed on the wake (clockwise (CR) vs. counterclockwise (CCR) vs. no rotation (NR)) and the inflow wind profiles (EBL with no wind veer vs. SBL with veering wind).

The rotational direction of a wind turbine rotor in an EBL without wind veer only exerts a minor influence on the wake behaviour, which is limited to the near wake. This minor impact is consistent with previous investigations by Vermeer et al. (2003), Shen et al. (2007), Sanderse (2009), Kumar et al. (2013), Hu et al. (2013), Yuan et al. (2014), and Mühle et al. (2017).

   In the presence of wind veer in an SBL, the rotational direction of a wind turbine, however, clearly impacts the streamwise elongation of the wake, its spanwise width, the velocity deficit in the near wake, and the wake deflection angle. As the operating
characteristics of upwind turbines, like their yaw with incoming flow, are already being adjusted to mitigate downwind impacts of wakes (Fleming et al., 2019), this work suggests that considering the direction of rotation could have benefits as well.

   A veering wind occurs in a SBL as long as the flow is not channeled. Veer reflects the vertical changes in interactions between the Coriolis force and friction in the absence of convection. Veer occurs on many nights both onshore (Walter et al., 2009; Rhodes and Lundquist, 2013; Sanchez Gomez and Lundquist, 2020) and offshore (Bodini et al., 2019a, b). According
to two years of meteorological tower measurements in Lubbock (Texas) (Walter et al., 2009), an SBL occurs in 52% of the measurements. Veering (of some degree) occurs in well over 70% of those SBL occurrences ($\approx 76\%$ in Walter et al. (2009) and $\approx 78\%$ in Sanchez Gomez and Lundquist (2020)). In this work, we apply a directional shear between 10 m and 115 m of $0.28\,^\circ\,\mathrm{m}^{-1}$, similar to what would be calculated based on standard values for the Ekman spiral in the atmosphere. This very large directional shear occurs only in a minority of considered nights of roughly 1% in Walter et al. (2009, Fig. 3). Based on
three months of lidar observations in north-central Iowa in Sanchez Gomez and Lundquist (2020, Fig. 7), a directional shear between 40 m and 120 m of $0.25\,^\circ\,\mathrm{m}^{-1}$, as it is the case in our SBL regime, occurs in 2% of the measurements. Comparing with 13 months of lidar observations off the coast of Massachusetts in Bodini et al. (2019a, Fig. 5), this very high wind veer occurs in 2% of the nights during summer and in 1% during winter. So in this study, to understand the possible significance of veer, we have explore a case with very strong veer.

In the majority of the cases with veering inflow, the directional shear has lower values in comparison to the SBL wind-turbine simulations presented in this work. For example, a direction shear of $\approx 0.08\,^\circ\,\mathrm{m}^{-1}$ occurs in 50% of the nights with veering wind in Walter et al. (2009). And yet, this work emphasizes the role of the interaction between inflow veer and wake rotation. When veer is present but smaller, we see an increasing impact of the rotational direction on the wake structure, being

aware that this is a nonlinear process approaching towards the non-veered situation if the directional shear approaches $0\,^\circ\,\mathrm{m}^{-1}$. Therefore, in the majority of the nights with veering inflow in Walter et al. (2009), Bodini et al. (2019a), and Sanchez Gomez and Lundquist (2020), the prevailing directional shear would have a larger impact on the meridional flow in the wake $v$ as in the case considered in this work (if the rotational velocity of the blades has not changed). Smaller veer would not only lead to

a stronger reduction of $v$ for a clockwise rotating rotor, but also to a reversal for small enough values of the directional shear. In the majority of the cases in these measurement campaigns, the wake difference between clockwise and counterclockwise rotating turbines would therefore be larger. Given the significant impact, the choice of rotational direction of wind-turbine blades in the Northern Hemisphere becomes more consequential.

Our simulations represent a canonical case, and so real-world conditions may modify results. The veering wind simulated

here is characteristic of a cloud free, nocturnal boundary-layer flow in the Northern Hemisphere. As the Coriolis force and the resulting Ekman spiral reverse direction in the Southern Hemisphere, the same effect should be prevalent in the Southern Hemisphere, however, with a reduction or reversion of the meridional wake component for counterclockwise rotating blades instead of clockwise rotating ones as it is the case in the Northern Hemisphere. Further, canonical Ekman spirals do not occur in every stably stratified ABL: conditions like cold air advection lead to a backing wind (Holton, 1973; Wallace and Hobbs, 2006),

which is a wind that rotates in the counterclockwise direction with increasing height in the Northern Hemisphere. A backing wind in the Northern Hemisphere interacting with both rotational directions will result in a different impact on the wake in comparison to a veering wind on the Northern Hemisphere. It should be comparable to the Southern Hemispheric situation described above. Further, the evolution of the stable boundary layer through the evening transition can lead to a variety of veer profiles, in which the veering inflow is limited to the lower rotor half (Rhodes and Lundquist, 2013; Lee and Lundquist, 2017) or

expands over the whole rotor (Abkar and Porté-Agel, 2016; Bromm et al., 2017; Churchfield and Sirnivas, 2018). But given the widespread occurrence of veer as noted by observational campaigns (Walter et al., 2009; Bodini et al., 2019b; Sanchez Gomez and Lundquist, 2020), as well as the interest in modifying wakes via active control of upwind turbines (Fleming et al., 2019), the work presented here can motivate further consideration of how inflow veer interacts with wind-turbine operations to affect downwind turbines.

*Author contributions.* All authors designed the idea. A. Englberger performed the simulations and prepared the manuscript with significant contributions from both co-authors.

*Competing interests.* The authors declare that they have no conflict of interest.

*Acknowledgements.* The authors gratefully acknowledge the Gauss Centre for Supercomputing e.V. (www.gauss-centre.eu) for funding this project by providing computing time on the GCS Supercomputer SuperMUC at Leibniz Supercomputing Centre (LRZ, www.lrz.de). This

work was authored [in part] by the National Renewable Energy Laboratory, operated by Alliance for Sustainable Energy, LLC, for the U.S. Department of Energy (DOE) under Contract No. DE-AC36-08GO28308. Funding provided by the U.S. Department of Energy Office of Energy Efficiency and Renewable Energy Wind Energy Technologies Office. The views expressed in the article do not necessarily represent the views of the DOE or the U.S. Government. The U.S. Government retains and the publisher, by accepting the article for publication, acknowledges that the U.S. Government retains a nonexclusive, paid-up, irrevocable, worldwide license to publish or reproduce the published form of this work, or allow others to do so, for U.S. Government purposes.

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
