# Peer review of "Does the rotational direction of a wind turbine impact the wake in a stably stratified atmospheric boundary layer?"

_Wind Energy Science, 2019_

## Referee Comment (RC1) · Anonymous Referee #1 · 10 Sep 2019

The authors used numerical simulations to study the effect of the rotational direction of a wind turbine on the wake subjected to the veering inflow. They considered different cases with and without veer with the clockwise and counterclockwise rotating turbine. The idea is of interest and relevant to the wind energy community. However, there are several issues in the simulations, modeling, and interpretations of the results. The main comments are as follows:

A. Simulations:

A1. The inflow conditions need to be added to the text, specifically, profiles of streamwise and lateral wind, temperature, turbulence intensities (in different directions) and

[Figure]

Reynolds shear stresses, as these parameters would affect the wake aerodynamics.

A2. It is mentioned that the turbulence intensity is the same in no-veer and veering cases. Is it turbulence intensity in the streamwise direction or the total turbulence level? As the other terms in the Reynolds stress tensor also contribute to the turbulent transport and the wake recovery, would it be enough to match the turbulence level in no-veer and veering cases?

A3. In veering inflow simulation, the wind direction changes with height, and it might also change with time. Is the wind direction fixed during the simulation? How the wind direction is kept constant during the simulation to avoid any yaw misalignment?

A4. What is the rotational speed of the turbine and how it is set during the simulations – with and without wind veer? The main characteristics of the turbine such as turbine RPM, turbine thrust and power coefficients should be added to the text for different cases.

A5. Based on the domain size, the blockage ratio is about 8% which is relatively large for simulations of a stand-alone wind turbine. It is recommended that the blockage ratio should be less than 2% to neglect its effect on wake development. Please clarify this point.

A6. The authors mentioned that the simulations are performed for 20 minutes and the results are averaged over the last 10 minute. Is 10min enough for the averaging? I believe a longer time should be considered for averaging especially for far wake region to make sure the statistics are well converged. I think the full recovery of the wake occurred at 16D might be related to limited simulation time.

B. Interpretation of the results:

B1. All the results are qualitative, and the paper suffers from the lack of quantitative comparisons. Velocity profiles at different distances from the turbine need to be plotted and added to the text to better quantified the effect of wake rotation.

B2. In the text, the analysis is only focused on the mean velocity (first-order statistics). Why only the mean velocity? How does the wake rotation affect the turbulence level or TKE behind the turbine in a stable regime?

B3. The authors use a fixed value for the wind veer (0.08 deg/m). What would be the effect of wind veer strength (or rotor size) on the results? As the effect of wind veer has been extensively studied before, it is expected that a more comprehensive investigation to be done.

B4. In page 9, line 3: the authors mentioned that: "this enhanced production of TKE due to the shear and wind veer . . .." The authors did not show any results about the TKE or TKE production in the text. It is not clear how they concluded that TKE production increases due to the veer.

B5. Following the previous comments, it would be useful that the authors show how the wake rotation can affect the TKE production in the wake.

C. Analytical modeling:

There are several major issues in the low-order model presented in the text:

C1. Eq(9): Why 0.3? Could you provide a physical justification for using 0.3? This value can be related to the induction factor of the turbine, but, in the current form, it is not justified.

C2. Eqs (13) and (14): This assumption should be verified or assessed by comparing the model to the simulation results. It is known that the lateral and vertical components of the velocity in the wake are different from the incoming wind.

C3. I could not follow how Eqs. (18-20) are obtained from Eqs. (15-17). Eqs. (15-17) are only a function of r. However, Eqs. (18-20) are a function of r and x.

Also, it should be explained how the magnitude of $\delta$ and $\gamma$ are obtained.

C4. Does the proposed model satisfy basic physics like mass and momentum conservation? This point needs to be clarified in the text.

---

## Referee Comment (RC2) · Anonymous Referee #2 · 29 Sep 2019

The manuscript presents research using LES to investigate the effect of wind veer and wind turbine rotation direction on the mean wake structure. A simplified superposition-based model combining variable veer and shear inflow with a Rankine vortex to represent the effect of wake rotation is able to recreate the general trends and sign of mean wake velocities.

Overall the study appears well designed to answer a basic question about the effect of wake rotation direction on the structure and rate of wake recovery for a horizontal axis three-bladed wind turbine. The structure of the manuscript is well organized, however there are a number of important details that need to be included regarding the description of the simulations and additional analysis to support interpretation of the results. The simple empirical model is shown to demonstrate some of the general physical influences on the structure of the wake. However, it is not apparent if the model may be useful beyond to demonstrate the interaction of wind veer and wake rotation on momentum recovery due to the number of empirical parameters that were adjusted to fit the simulations without physical explanation or generalization.

It is my judgement that Major Revisions are required before the paper can be considered for publication.

Specific comments:

1) Title: Although titles in the form of a question can generate curiosity, I find it more impactful to simply state the main hypothesis of the study, such as for example: Parametric study on the effects of wind veer and wind turbine rotation direction on the structure and recovery rate of the mean wake.

2) Introduction: There is a mixture of future and present tense used throughout. Please be consistent. Careful editing of the text is needed throughout the manuscript to ensure clarity of the presentation, particularly clear physical explanations and accurate word choice. For example, in the first paragraph of the Introduction. "The diurnal cycle is driven by shortwave heating during day and radiative cooling at night." Both are radiative processes, shortwave solar heating for the land surface and longwave radiative cooling. This is followed by a statement about forces acting on velocity: "The interaction between the Coriolis force acting on the velocity components. . ."

3) Pg 2: the review of previous research could be more descriptive. For example what methods are employed in the various studies to investigate stability dependence on the ABL? Are there any important features that have been regarding the wake of wind turbines operating in stable and convective boundary layers compared to neutral?

4) Pg 2, Line 24: I believe Hui Hu's group at Iowa State published a number of papers on wind tunnel experiments to investigate the counter rotating, dual-rotor turbine, starting around 2014. It would be good to provide a comprehensive review for studies considering rotor rotation given the focus.

5) Fig. 1: note that rotor blades must be rotating to generate axial thrust, however this affect may be modeled using a drag disk, not including the effect of conservation of angular momentum. If I understand the text, this is the approach represented by subfigures 2c and 2d, but the language used is a bit confusing.

6) Section 2, Numerical Model Framework: The details of the model pertinent to the simulations conducted in this study should be explained in detail. It is appropriate to point to a former paper if validation is reported in support of this study. It is not clear if the paper cited provides these details. Nevertheless, the details of the specific model implementation in this study should be explained.

7) Pg. 4: Equation 2 appears incomplete. It may be easier to follow if first the description of Equations 1-3 are provided followed by explanation of the terms and definition of each variable. Details about the implementation of the turbulence closure and turbine actuator model should be included. Are the implementations similar to Wu and Porte-Agel, or C. Archer?

8) Please explain how the inflow velocity profile is imposed in the simulations. How is a proper turbulence profile established for each case?

9) Was the TKE model tested or calibrated to ensure the correct balance and transport is modeled in the wake region under the different flow and turbine operating conditions?

10) Pg. 5: How is the turbine model scaled? Based on geometry only? Or also considering operational characteristics? What are the details?

11) Pg. 5: Please include a schematic and complete description of the simulation set-up, including size of the domain. What is the fraction of the channel cross-section occupied by the rotor disk? Is the grid spacing uniform? How many grid points represent the rotor? Are the turbine forces applied uniformly or based on some scheme related to the blade geometry and aerodynamics?

12) Pg. 6: What is the corresponding land surface roughness, thermal stability, and latitudinal location represented by the imposed profiles uses in the simulations.

13) Regarding the turbine operation, what tip speed ratio and thrust coefficient are modeled? Are these typical operating conditions?

14) Please check throughout and change "turbulent intensity" to "turbulence intensity".

15) Table 2: It is not clear from the later discussion of the results if both the 60 deg and 90 deg sectors are used in the analysis?

16) Pg. 7: The use of index notation for velocity commonly refers to components of the vector, which may be confused with the (x, y, z) coordinates intended here.

17) Fig. 3: Labels on the contours are difficult to read due to crowding and may be improved for clarity.

18) Consider plotting distributions of turbulence intensity in the wake and provide any insight on the patterns seen for the distributions of the mean wake velocity.

19) Section 3: Check the use of $z$, $z\_h$, $k\_h$, and $k\_*$. There appears to be some inconstancy that may lead to confusion.

20) Pg 9, Lines 3-5: Would be useful to show plots of TKE to support the point about enhanced production effects on wake recovery.

21) Pg 9, Line 10: Check NV_VR, should be NV_NR.

22) Figs 4 and 5: Quantification of the y-direction momentum budget could be used to support the assertion that the inflow veer and direction of wake rotation either partially cancel out or enhance wake deflection in the upper and lower portions of the wake.

23) Are there any data to support the simulation results of the wake you observe in the

y-z plane?

24) Fig. 7: Please explain the relatively stronger CW rotation in subfigure 7g compared to 7d and 7h. Why does the wake rotation switch from CCW sense to CW?

25) You might consider quantifying the Circulation within the rotor region to quantitatively compare the cases.

26) Fig. 8: Please comment on the relatively strong counter rotating structures observed between the rotor and the corners of the domain. Might these be weaker if the domain were enlarged? How might this affect the comparison with the simple model?

27) Sect 4.1, Model Development: It would be useful to relate the models, particularly for the axial component of velocity to existing analytical wake models. For example, what is the physical meaning of the 0.3 used in Eqn. 9? What are the axial and tangential induction factors used and are they appropriate for the turbine being modeled?

28) Be sure to define all variables and subscripts used throughout the model development. For example, _RV, _M, _fad are not explicitly defined.

29) Eqns 18 – 20, Given the premise that at least some of the flow conditions are caused by thermal stability, how could stability be included in the model, as the normal and shear components of turbulence fluxes will significantly affect the wake evolution.

30) How are the values of the parameters x_rec, x_fad, gamma, and delta chosen? Is it possible to estimate them without the flow data from simulations?

31) Pg 16, Line 14: Please check the single sentence paragraph.

32) Fig 9: It would be easier to compare the model with the LES output if they were plotted together and instead separate the CCW and CW cases. Note that the legend above the figure does not coincide with the figure description below for the vertical velocity plots.

33) Please check that average velocity is defined and used consistently. Note in Fig 9,

an overbar is used, but not in the figure description or elsewhere. Are the plotted data based on the 60 deg or 90 deg sectors defined in Table 2?

34) Consider comparing the results to field data. In particular, there are a number of case experiments that have used lidar to measure the wake. Even if not the data are not directly compatible due to different turbine models or operating conditions, it would be useful to see if measurements see the same trends presented in this study.

35) Section 5: It is not clear if thermal stability was actually modeled, or only the Ekman spiral effect on the wind direction. The language used throughout the manuscript may contribute to confusion about what physics are considered in this study. This can be avoided by using explicitly clear language.

36) Statements in the Conclusion that provide interpretation with various supporting references should be moved up to the Discussion. The Conclusion should focus on the main outcomes of the present study.

---

## Author Comment (AC1) · 31 Oct 2019

**Answers to Anonymous Referee 1 on the paper 'Does the rotational direction of a wind turbine impact the wake in a stably stratified atmospheric boundary layer?'**

Englberger et al.

**A.Simulations**

**A1**

The inflow conditions need to be added to the text, specifically, profiles of stream wise and lateral wind, temperature, turbulence intensities (in different directions) and Reynolds shear stresses, as these parameters would affect the wake aerodynamics.

The profiles of $u$ and $v$ are given in Eqs. 4 and 5. We forget to mention the potential temperature profile, we added it now as: 'The potential temperature is

$$\Theta_{BL}(z) = \frac{3\,K}{200\,m} z \tag{1}$$

in the lowest 200 m and 303 K above. '

As they are idealized profiles we think it is not mandatory to show them in a plot. The turbulence intensity is discussed in detail in Englberger and Dörnbrack (2018a), we cited the reference in the paper.

The Reynolds stress tensor terms $u'u'$, $w'w'$, and $u'w'$ are exactly the same in $NV$ and $V$ simulations. As we discuss the differences in the wake in this paper, which only depend on the differences between $V$ and $NV$ and not on individual values, we decided to add a figure in the paper showing the relevant differences of lateral wind and Reynolds stresses:

   a  the incoming wind $v$ and the incoming wind direction $\phi$ in Fig. 2(a)

   b  the Reynolds stress tensor terms $u'v'$ and $v'v'$ (and $v'w'$ with only minor differences) in Fig. 2(b)

Representing only the differences between $V$ and $NV$ is further helpful to understand the differences in the flow rotation plot outside the rotor (Fig. 10 and 11).

It is included in the text as:

   a  'The differences in the inflow conditions between veered and non-veered simulations are presented in Fig. 2(a) for the spanwise velocity $v$ and the wind direction with respect to $270\,^{\circ}$ as $V$ - $NV$.'

   b  'The ABL flow (Eqs. 4-8) in combination with the impressed turbulence of a stably stratified regime result in exactly the same Reynolds stress tensor terms of $u'u'$, $w'w$, and $u'w'$ in $V$ and $NV$ whereas there are differences in $v'v'$, $u'v'$, and

$v'w'$ (Fig. 2(b)). In the height of the rotor, $u'v'$ is symmetric with respect to hub height and both $u'v'$ and $v'v'$ increase approaching the blade tip, whereas $v'w'$ is marginal.'

[Figure]

**Figure 1.** Differences in the initial conditions between $V$ and $NV$ for the spanwise velocity $v$ and the incoming wind direction $\phi$, whereas the differences are related to 270°, in (a) and the Reynolds stress tensor terms $v'v'$, $u'v'$ and $v'w'$ in (b).

**A2**

It is mentioned that the turbulence intensity is the same in no-veer and veering cases. Is it turbulence intensity in the streamwise direction or the total turbulence level? As the other terms in the Reynolds stress tensor also contribute to the turbulent transport and the wake recovery, would it be enough to match the turbulence level in no-veer and veering cases?
See A1. We additionally discussed and plotted the Reynolds stress tensor terms.

**A3**

In veering inflow simulation, the wind direction changes with height, and it might also change with time. Is the wind direction fixed during the simulation?
In the simulations, the wind direction changes only with height, they are fixed with time.
How the wind direction is kept constant during the simulation to avoid any yaw misalignment?
We verify a constant wind direction by applying the Eqs. 4, 5, and 7 of $u_{BL}(z)$, $v_{BL}(z)$, and $w_{BL}(z)$. This results in $v(hub) = 0$ all the time. The applied parametrization of Englberger and Dörnbrack (2018b) verifies no positive or negative tendency of $v$ at each height for the considered 10 min averaging time. Therefore, the flow is directly perpendicular to the nacelle at hub height and symmetric in the lower and the upper rotor part.

**A4**

What is the rotational speed of the turbine and how it is set during the simulations – with and without wind veer? The main characteristics of the turbine such as turbine RPM, turbine thrust and power coefficients should be added to the text for different

cases.

The rotational speed of the turbine is fixed at 7 rpm in all simulations. We added the value in the description of the wind-turbine parametrization. 'Further, the rotation frequency $\Omega$ is set to 7 rpm.'

As we apply the BEM method, we do not have a constant $c_T$ or $c_P$ value. However, we cited the turbine, we used for the parametrization of the BEM method.

**A5**

Based on the domain size, the blockage ratio is about 8% which is relatively large for simulations of a stand-alone wind turbine. It is recommended that the blockage ratio should be less than 2% to neglect its effect on wake development. Please clarify this point.

As blockage is relevant for confined simulations, it should not be relevant for this work as we present open boundary condition simulations.

**A6**

The authors mentioned that the simulations are performed for 20 minutes and the results are averaged over the last 10 minute. Is 10min enough for the averaging? I believe a longer time should be considered for averaging especially for far wake region to make sure the statistics are well converged.

In Englberger and Dörnbrack (2018a) we investigated the effect of the time used for averaging resulting that under moderately and strongly stably stratified (SBL and MBL) conditions, the 10 min, 30 min and 50 min averages result in an almost identical turbulent intensity profile. In contrast, it was found that under convective and near-neutral (CBL and EBL) stratifications, a longer averaging period of at least 30 min is necessary. Therefore, the considered SBL case resulting from the same diurnal cycle precursor simulation from Englberger and Dörnbrack (2018a) shows that an averaging time of 10 min is enough.

I think the full recovery of the wake occurred at 16D might be related to limited simulation time.

We are not sure why the reviewer suggests a full wake recovery at $16\,\mathrm{D}$, according to Figs. 4, 5, and 6 this is not the case for $x < 20\,\mathrm{D}$.

**B. Interpretation of the results**

We agree with the Reviewer and included vertical and spanwise profiles including velocity and turbulent intensity information at $10\,\mathrm{D}$ downstream to allow a more quantitative comparison between the three considered veered cases.

**B1**

All the results are qualitative, and the paper suffers from the lack of quantitative comparisons. Velocity profiles at different distances from the turbine need to be plotted and added to the text to better quantified the effect of wake rotation.

We added vertical and spanwise profiles presenting the difference in velocity deficit $\Delta V D$ between each two of the discussed

simulations in the paper (Figs. 2 (a, c, e, g) and 3 (a, c, e, g)). We decide not to include a plot showing the exact values of vertical and horizontal profiles of $u$ for all six simulations (as tested in Fig. 4), as showing $\Delta VR$ is much more representative, the difference is much easier to see and further the individual figures are not of major interest as it is an idealized study.

**B2**

In the text, the analysis is only focused on the mean velocity (first-order statistics). Why only the mean velocity? How does the wake rotation affect the turbulence level or TKE behind the turbine in a stable regime?

See B1 part and Figs. 2 (b, d, f, h) and 3 (b, d, f, h).

**B3**

The authors use a fixed value for the wind veer (0.08 deg/m). What would be the effect of wind veer strength (or rotor size) on the results? As the effect of wind veer has been extensively studied before, it is expected that a more comprehensive investigation to be done.

The impact of the strength of wind veer and also the rotor position affected by veer (upper or lower rotor half or full rotor as in this study) is a very interesting topic, and is discussed in detail in two further publications, one will be published in the next days Englberger and Lundquist (2019) and on will be submitted within in the next weeks Englberger et al. (2019). The wind veer strength has an impact on the difference between wake veer and inflow wind veer. This difference decreases if the vertical gradient of the veering wind increases (Englberger and Lundquist, 2019). Further, a larger vertical gradient of the veering wind decreases the wake difference induced by the rotational direction (Englberger et al., 2019). We did not add it in this work, as it is a rather complex issue, with additional parameters impacting the wake difference.

The rotor size is not changed. However, changing the amount of deg/m should be comparable to a constant wind veer value while changing the rotor diameter. Both would result in the same amount of wind veer over the whole rotor.

**B4**

In page 9, line 3: the authors mentioned that: "this enhanced production of TKE due to the shear and wind veer . . .." The authors did not show any results about the TKE or TKE production in the text. It is not clear how they concluded that TKE production increases due to the veer.

We added TKE now to the paper (see B2).

**B5**

Following the previous comments, it would be useful that the authors show how the wake rotation can affect the TKE production in the wake.

We added TKE now to the paper (see B2).

[Figure]

**Figure 2.** Vertical and spanwise profiles of $\Delta VD$ in (a), (c), (e), (g) and $\Delta I$ in (b), (d), (f), (h) at $10\,\mathrm{D}$. The spanwise profiles are plotted at 75 m (orange frame, (g), (h)), 100 m (hub height) (green frame, (e) (f)), and 125 m (purple frame (c), (d)). In panels (a) and (b), the coloured lines indicate the altitudes analysed in (c) - (h). Considering a comparison of two simulations A and B (see legend: A vs B), $\Delta VD$ is calculated as the difference between simulation B and simulation A $VD(B) - VD(A)$ and $\Delta I$ as the difference between simulation A and simulation B $I(A) - I(B)$. Therefore, $\Delta VD > 0$ and likewise $\Delta I > 0$ represent a more rapid wake recovery of simulation A in comparison to simulation B, related to a higher turbulence intensity level.

[Figure]

**Figure 3.** Vertical and spanwise profiles of $\Delta VD$ in (a), (c), (e), (g) and $\Delta I$ in (b), (d), (f), (h) at $10\,\mathrm{D}$. The spanwise profiles are plotted at $75\,\mathrm{m}$ (orange frame, (g), (h)), $100\,\mathrm{m}$ (hub height) (green frame, (e) (f)), and $125\,\mathrm{m}$ (purple frame (c), (d)). In panels (a) and (b), the coloured lines indicate the altitudes analysed in (c) - (h). Considering a comparison of two simulations A and B (see legend: A vs B), $\Delta VD$ is calculated as the difference between simulation B and simulation A $VD(B) - VD(A)$ and $\Delta I$ as the difference between simulation A and simulation B $I(A) - I(B)$. Therefore, $\Delta VD > 0$ and likewise $\Delta I > 0$ represent a more rapid wake recovery of simulation A in comparison to simulation B, related to a higher turbulence intensity level.

[Figure]

**Figure 4.** Vertical profiles of the velocity deficit in a, b and the turbulent intensity in c, d and the corresponding spanwise profiles in e, f, g, and h at two downstream distances of 5 D (first row) and of 10 D (second row), each through the centre of the rotor disc.

**C. Analytical modeling**

**C1**

Eq(9): Why 0.3? Could you provide a physical justification for using 0.3? This value can be related to the induction factor of the turbine, but, in the current form, it is not justified.

It is motivated on page 15 line 12 (original manuscript). According to Fig. 4, the maximum value of the velocity deficit is roughly 0.7, resulting in the factor 0.3 used in Eq. 9.

The calculation of the velocity deficit

$$VD_{i,j,k} \equiv \frac{\overline{u_{1,j,k}} - \overline{u_{i,j,k}}}{\overline{u_{1,j,k}}}. \tag{2}$$

is directly related to the axial induction factor

$$a = \frac{V_\infty - V_R}{V_\infty} \tag{3}$$

following Manwell et al. (2002) and Hansen (2008), resulting in a value of 0.3=1-a in the calculation of the axial force as used in momentum theory.

We added the comparison with the axial induction factor and momentum theory in the paper as:

'We apply a fraction of 0.3, which can be related to $VD_{max} = 0.7$ of the rotating disc simulations in Fig. 4 and consequently to an axial induction factor $a$ of 0.7, resulting in a fraction of $0.3 = (1\text{-}a)$, as it would be the case by calculating the axial force with momentum theory Manwell et al. (2002); Hansen (2008).'

Further, we modified the definition of $u_M$ to make it more straight-forward. We changed from

$$u_M(r, x_{pos}) = 1.3 \cdot u_{BL} \left( \frac{x_{pos}}{x_{rec}} \right)^\gamma \tag{4}$$

to

$$u_M(r, x_{pos}) = u_{BL} - 0.3 \cdot u_{BL} \left( \frac{x_{rec} - x_{pos}}{x_{rec}} \right). \tag{5}$$

This results in an even better fit of $\overline{u}$ in Fig. 6 in comparison to 5, however, with no relevance for the important components of $v$ and $w$.

**C2**

Eqs (13) and (14): This assumption should be verified or assessed by comparing the model to the simulation results. It is known that the lateral and vertical components of the velocity in the wake are different from the incoming wind.

The comparison is presented in Fig. 12.

The difference in the wake from the incoming wind is considered by including $v_{WT}$ and $w_{WT}$. We added in the text: 'From a linear superposition of the boundary layer flow with the wind-turbine induced forces on the flow field results for the developed

[Figure]

**Figure 5.** Sector averages $\overline{u}$ in (a), $\overline{v}$ in (c), and $\overline{w}$ in (e) for the veered cases V_CCW and V_CW. $\overline{u}$ and $\overline{v}$ show the top-tip and bottom-tip sectors, $\overline{w}$ the right (solid line) and left (dashed line) sectors. The corresponding model components are shown in (b) for $\overline{u}$, (d) for $\overline{v}$, and (f) for $\overline{w}$.

simplified wake model ($M$):'
Further, we are aware of the fact that our simplified model does not include any non-linear entrainment processes etc., however, as its aim is only to explain the differences in the $v$ and $w$ components, this simplification is considered to be appropriate.

**C3**

I could not follow how Eqs. (18-20) are obtained from Eqs. (15-17). Eqs. (15-17) are only a function of r. However, Eqs. (18-20) are a function of r and x.
Eqs. (15-17) (no 16-18) only depend on the radial position. Now, the downstream behaviour is included in Eqs. (18-20) (now

[Figure]

**Figure 6.** Sector averages $\overline{u}$ in (a), $\overline{v}$ in (c), and $\overline{w}$ in (e) for the veered cases V_CCW and V_CW. $\overline{u}$ and $\overline{v}$ show the top-tip and bottom-tip sectors, $\overline{w}$ the right (solid line) and left (dashed line) sectors. The corresponding model components are shown in (b) for $\overline{u}$, (d) for $\overline{v}$, and (f) for $\overline{w}$.

19-21). Here, approaching downstream, the $u$, $v$, and $w$ components increase due to wake recovery. The $v$ and $w$ components are also affected by the rotational direction of the ambient flow. The way of including the downstream position is explained by 1. and 2. on page 19. Therefore, now Eqs. (19-21) are a function of the radial position in relation to the rotor center and likewise on the downstream distance from the rotor. Therefore, with Eqs. (19-21) an assumption at each waked grid point is possible.

Also, it should be explained how the magnitude of $\delta$ and $\gamma$ are obtained.

The magnitude of $\gamma$ and $\delta$ result from archiving the best possible agreement between the model results and the simulation results (e.g. see Fig. 12). We added: 'The values of $\gamma$ and $\delta$ are determined by empirical fitting.' We consider empirical fitting appropriate, as the values of $\delta$ and $\gamma$ only modify the amount of $v$ and $w$ downwind, however, not the sign of both.

**C4**

Does the proposed model satisfy basic physics like mass and momentum conservation? This point needs to be clarified in the text.

Conservation of mass: Considering Eqs. 18-20,

$$u_M(r, x_{pos}) = u_{BL} - 0.3 \cdot u_{BL} \left( \frac{x_{rec} - x_{pos}}{x_{rec}} \right) \tag{6}$$

$$v_M(r, x_{pos}) = \left( v_{BL} \pm (\pm \omega r \frac{r}{R} cos(\Theta)) \left( \frac{1}{exp \frac{x_{pos}}{x_{fad}}} \right)^{\delta} \right) \left( \frac{x_{pos}}{x_{rec}} \right)^{\gamma} \tag{7}$$

$$w_M(r, x_{pos}) = \pm (\mp \omega r \frac{r}{R} sin(\Theta)) \cdot \left( \frac{1}{exp \frac{x_{pos}}{x_{fad}}} \right)^{\delta} \cdot \left( \frac{x_{pos}}{x_{rec}} \right)^{\gamma}, \tag{8}$$

and the continuity equation. Considering a certain position $x_{pos}$, from Eq. 6 follows $\frac{\partial u}{\partial x} = 0$. Further, considering $r^2 = y^2 + z^2$, $\frac{\partial v}{\partial y} \approx constant * y$ and $\frac{\partial w}{\partial z} \approx constant * z$. Therefore, form the continuity equation follows y+z=0. This can be interpreted as a conservation of mass on concentric circles around hub height. This is in agreement with the development of the model, which was assumed to be radially symmetric to the nacelle. Considering complex entrainment processes, we think the mass has not to be conserved at specific downstream regions, as it is the case over the whole domain.

Conservation of momentum:

This is verified as the model is constant in time and, no temporal changings are expected. Further, the model is linear, therefore no advection, only Coriolis force and pressure gradient force (no pressure change) are present.

**References**

Englberger, A. and Dörnbrack, A.: Impact of the diurnal cycle of the atmospheric boundary layer on wind-turbine wakes: a numerical modelling study, Boundary-layer meteorology, 166, 423–448, https://doi.org/10.1007/s10546-017-0309-3, 2018a.

Englberger, A. and Dörnbrack, A.: A Numerically Efficient Parametrization of Turbulent Wind-Turbine Flows for Different Thermal Stratifications, Boundary-layer meteorology, 169, 505–536, https://doi.org/10.1007/s10546-018-0377-z, 2018b.

Englberger, A. and Lundquist, J. K.: How does inflow veer affect the veer of a wind turbine wake?, in: North American Wind Energy Academy 2019 Symposium, Virginia Tech, 2019.

Englberger, A., Lundquist, J., and Dörnbrack, A.: Should wind turbines rotate in the opposite direction?, to be submitted, 2019.

Hansen, M. O.: Aerodynamics of wind turbines, vol. 2, Earthscan, London and Sterling, UK and USA, 181 pp, 2008.

Manwell, J., McGowan, J., and Roger, A.: Wind Energy Explained: Theory, Design and Application, Wiley: New York, NY, USA, 134 pp, 2002.

---

## Author Comment (AC2) · 31 Oct 2019

**Answers to Anonymous Referee 2 on the paper 'Does the rotational direction of a wind turbine impact the wake in a stably stratified atmospheric boundary layer?'**

Englberger et al.

**Specific comments**

Title: Although titles in the form of a question can generate curiosity, I find it more impactful to simply state the main hypothesis of the study, such as for example: Parametric study on the effects of wind veer and wind turbine rotation direction on the structure and recovery rate of the mean wake.

We think it is a matter of style. In this work, we think a question is suitable as it is answered with a clear yes in the Conclusion.

Introduction: There is a mixture of future and present tense used throughout. Please be consistent. Careful editing of the text is needed throughout the manuscript to ensure clarity of the presentation, particularly clear physical explanations and accurate word choice.

We reread the text taking care of future and present tense.

For example, in the first paragraph of the Introduction. "The diurnal cycle is driven by shortwave heating during day and radiative cooling at night." Both are radiative processes, shortwave solar heating for the land surface and longwave radiative cooling. This is followed by a statement about forces acting on velocity: "The interaction between the Coriolis force acting on the velocity components. . ."

We added longwave in front of radiative cooling at night.

Pg 2: the review of previous research could be more descriptive. For example what methods are employed in the various studies to investigate stability dependence on the ABL? Are there any important features that have been regarding the wake of wind turbines operating in stable and convective boundary layers compared to neutral?

We added the skewness as additional feature in the SBL: The stability-dependent wind and turbulence conditions determine the entrainment of energy and momentum into the wake region and the resulting wake structure, with fast eroding wakes in convective conditions and wakes persisting further downwind in stably stratified conditions which are additionally characterized by a

skewed wake (Abkar et al., 2016; Vollmer et al., 2017; Englberger and Dörnbrack, 2018). This stability dependence has been investigated in numerical simulations for the SBL (Aitken et al., 2014; Bhaganagar and Debnath, 2014, 2015; Dörenkämper et al., 2015), the CBL (Mirocha et al., 2014), both of them (Abkar and Porté-Agel, 2014; Vollmer et al., 2017), or the complete diurnal cycle (Abkar et al., 2016; Englberger and Dörnbrack, 2018).

Pg 2, Line 24: I believe Hui Hu's group at Iowa State published a number of papers on wind tunnel experiments to investigate the counter rotating, dual-rotor turbine, starting around 2014. It would be good to provide a comprehensive review for studies considering rotor rotation given the focus.

This paper was cited as Wei et al. 2014.

Fig. 1: note that rotor blades must be rotating to generate axial thrust, however this affect may be modeled using a drag disk, not including the effect of conservation of angular momentum. If I understand the text, this is the approach represented by subfigures 2c and 2d, but the language used is a bit confusing. In Fig. 1c and 1d, there is only an axial force acting on the wake, no tangential one. Whereas in 1a, b, e, f there is also a tangential force. We just mentioned this in the introduction to present the problem. In chapter 2.2 it is explained in detail by the component $\beta_{\boldsymbol{v}} \in \{ 0, -1, 1\}$.

'$\mathbf{F}_{WT}$ corresponds to the turbine-induced force and the scalar prefactors $\beta_{\mathbf{v}} = (1, \beta_v, \beta_w)$ to the rotation of the rotor. The Coriolis force is represented by the angular velocity vector of the Earth's rotation.'

and further:

'... In these simulations, only the prefactors $\beta_v$ and $\beta_w$ in Eq. 1 differ. A clockwise wake rotation is defined by $\beta_v = -1$ and $\beta_w = 1$, a counterclockwise wake rotation by $\beta_v = 1$ and $\beta_w = -1$, and no rotation by $\beta_v = 0$ and $\beta_w = 0$, with $\beta_u = 1$ in each simulation. ...'

Section 2, Numerical Model Framework: The details of the model pertinent to the simulations conducted in this study should be explained in detail. It is appropriate to point to a former paper if validation is reported in support of this study. It is not clear if the paper cited provides these details. Nevertheless, the details of the specific model implementation in this study should be explained.

The details are given in Eq. 1-3. Here we changed the environmental state e.g. $v_e$ to the boundary-layer state $v_{BL}$ to make it clearer. All terms are defined below.

Further, we expanded the explanation of the wind-turbine parametrization in 2.2.

Pg. 4: Equation 2 appears incomplete. It may be easier to follow if first the description of Equations 1-3 are provided followed by explanation of the terms and definition of each variable.

We added the missing term. Now Eq. 2 says:

$\frac{d\Theta'}{dt} = -\mathbf{v}\nabla\Theta_e + \mathcal{H}$

An explanation of each term is given in the paper following Eg. 2.

Details about the implementation of the turbulence closure and turbine actuator model should be included. Are the implementations similar to Wu and Porte-Agel, or C. Archer?

- – Turbulence closure:

  Details about the implementation of turbulence closure can be found in Schmidt and Schumann, 1989 and Margolin et al. 1999 (cited in the paper)

- – Actuator model:

  We expanded the explanation of the actuator model to:

  'The axial $\mathbf{F}_x$ and tangential $\mathbf{F}_\Theta$ (Eqs. 9 and 10) turbine-induced forces ($\mathbf{F}_{WT} = \mathbf{F}_x + \mathbf{F}_\Theta$) in Eq. 1 are parametrized with the BEM method as rotating actuator disc including a nacelle and excluding the tower.

$$\left. |F_x| \right|_{x_0,y,z} = \frac{1}{2}\rho_0 \frac{Bc}{2\pi r_{x_0,y,z}}(c_L \cos\Phi + c_D \sin\Phi)$$
$$\times A_{x_0,y,z} \frac{u_{x_\infty,y,z}^2 (1-a)^2}{\sin^2\Phi} \tag{1}$$

$$\left. |F_\Theta| \right|_{x_0,y,z} = \frac{1}{2}\rho_0 \frac{Bc}{2\pi r_{x_0,y,z}}(c_L \sin\Phi - c_D \cos\Phi)$$
$$\times A_{x_0,y,z} \frac{u_{x_\infty,y,z}(1-a)\Omega r_{x_0,y,z}(1+a')}{\sin\Phi\cos\Phi}. \tag{2}$$

  Here, the centre of the rotor in $x$-direction is defined by the grid-point coordinate $x_0$. $B$ represents the number of blades, $c$ is the chord length of the blade, $c_L$ is the lift coefficient, $c_D$ is the drag coefficient, $\Phi$ is the angle between the plane of rotation and the relative streamwise velocity, and $a'$ is the tangential induction factor. Following Hansen (2008), we calculate $a$ and $a'$ by an iterative procedure from the airfoil data. For the airfoil data, the 10 MW reference wind turbine from DTU (Bak et al., 2013) is applied, whereas the radius of the rotor as well as the chord length of the blades are scaled to a rotor with a diameter of 100 m. The upstream velocity $u_{x_\infty,y,z}$ is taken at the first upstream grid point in the $x$-direction and the corresponding $y$ and $z$ coordinates. With the exception of $\rho_0$ and $B$, all other parameters appearing in Eqs. 1 and 2 depend on the radius $r_{x_0,y,z}$ and vary spatially. Further, the rotation frequency $\Omega$ is set to 7 rpm. A more detailed description of the wind-turbine parametrization and the applied smearing of the forces, as well as all values used in the wind-turbine parametrization can be found in Englberger and Dörnbrack (2017, parametrization B).'

– Comparison:

Porte-Agel: LES model and actuator disc model

C. Archer: Wind Turbine and Turbulence Simulator and actuator line model

Our paper: LES model EULAG and actuator disc model → Comparable to Porte-Agel, however, different numerical model and different wind turbine represented in the simulations. Not directly comparable to C. Archer. A comparison of our model with others can be found in Englberger and Dörnbrack (2017), as cited in the paper.

Please explain how the inflow velocity profile is imposed in the simulations. How is a proper turbulence profile established for each case?

We expanded the explanation in the paper with: 'The parametrization considers three 3 D wind fields ($u$, $v$, and $w$), resulting from a neutral ABL precursor simulation, which are modified by a stratification-dependent weighting, resulting from the SBL state of a diurnal cycle precursor simulation.'

We added a detailed description of the Reynolds stress tensor terms, especially regarding the differences resulting between the veered and the non-veered simulations and included an additional Figure as: 'The ABL flow (Eqs. 4-8) in combination with the impressed turbulence of a stably stratified regime result in exactly the same Reynolds stress tensor terms of $u'u'$, $w'w$, and $u'w'$ in $V$ and $NV$ whereas there are differences in $v'v'$, $'u'v'$, and $v'w'$ (Fig. 2(b)). In the height of the rotor, $u'v'$ is symmetric with respect to hub height and both $u'v'$ and $v'v'$ increase approaching the blade tip, whereas $v'w$ is marginal.'

[Figure]

**Figure 1.** Differences in the initial conditions between $V$ and $NV$ for the spanwise velocity $v$ and the incoming wind direction $\phi$, whereas the differences are related to 270°, in (a) and the Reynolds stress tensor terms $v'v'$, $u'v'$ and $v'w'$ in (b).

Was the TKE model tested or calibrated to ensure the correct balance and transport is modeled in the wake region under the different flow and turbine operating conditions?

We use a standard TKE closure for LES. This means it is not calibrated for a specific problem. The coefficients result from Schmidt and Schumann (1989).

Pg. 5: How is the turbine model scaled? Based on geometry only? Or also considering operational characteristics? What are the details?

It is scaled based on the geometry. To our opinion this is valid considering the application of an acutator disc model with a grid spacing of 5 m. Further, the results of this study only depend on the rotational direction and are independent of the proposed wind turbine detailed blade characteristics, the only impact factor is the rotor diameter and the hub height interacting with the incoming wind profile.

Pg. 5: Please include a schematic and complete description of the simulation set-up, including size of the domain. What is the fraction of the channel cross-section occupied by the rotor disk? Is the grid spacing uniform? How many grid points represent the rotor? Are the turbine forces applied uniformly or based on some scheme related to the blade geometry and aerodynamics?

We included missing information.

  – Size of domain is given with 512 x 64 x 64 grid points and a resolution of 5 m.

  – The fraction of the channel cross-section is 100 m/(64*5 m) = 0.31 (See rotor diameter and domain size in 2.2).

  – Yes, the grid spacing is uniform (see in the paper: with a horizontal and a vertical resolution of 5 m). We added uniform to the text.

  – We applied the blade element momentum method (BEM) for the parametrization of the axial and tangential forces. The forces in the BEM have a radial dependency and further depend on the inflow velocity profile. Therefore, they are not uniform and depend on blade geometry and wind conditions. We expanded the description of the actuator parametrization in the text to make it clearer. (See 7)

Pg. 6: What is the corresponding land surface roughness, thermal stability, and latitudinal location represented by the imposed profiles uses in the simulations.

We apply a drag coefficient of 0.1 in our simulations. It represents homogeneous terrain and should be comparable to a surface

roughness of the order of O(0.01m).

Thermal stability: We missed it in the paper and included it as: 'The potential temperature is

$$\Theta_{BL}(z) = \frac{3\,K}{200\,m} z \tag{3}$$

in the lowest 200 m and 303 K above.'

Latitudinal location: We listed the Coriolis parameter $f = 1.0 \times 10^{-4}$ s$^{-1}$. With $f = 2\Omega sin(\phi)$, a latitude on earth of $\phi \approx 43°$ arises.

Regarding the turbine operation, what tip speed ratio and thrust coefficient are modeled? Are these typical operating conditions?

We included the information of 7 r.p.m. in the paper. This results with the upstream profile of Eq. 4 in a tip speed ratio of 0.58. As we apply the BEM approach, we do not have a single $c_T$ value as in momentum theory. Therefore, we apply the values of the 10 MW reference wind turbine from DTU and cited them as Bak et al. (2013) in the paper.

Please check throughout and change "turbulent intensity" to "turbulence intensity".

We changed it.

**15**

Table 2: It is not clear from the later discussion of the results if both the 60 deg and 90 deg sectors are used in the analysis?

Only the results of 60° sectors are used. This is clarified in 2.3 in the paper: 'As the features are more distinctive for 60 ° and 25 m $\leq$ r $\leq$ 50 m sectors, these values are applied for the derivation of the sector characteristics in the following.'

**16**

Pg. 7: The use of index notation for velocity commonly refers to components of the vector, which may be confused with the (x, y, z) coordinates intended here.

'The indices of the grid points are denoted by $i = 1 \ldots n$, $j = 1 \ldots m$ and $k = 1 \ldots l$ in the $x$, $y$ and $z$ directions, respectively.' The use of discretized values in this way is standard for numerical purposes. This should not be linked to the Einstein notation.

**17**

Fig. 3: Labels on the contours are difficult to read due to crowding and may be improved for clarity.

We improved it.

**18**

Consider plotting distributions of turbulence intensity in the wake and provide any insight on the patterns seen for the distributions of the mean wake velocity.

We added vertical and spanwise profiles presenting the difference in velocity deficit $\Delta VD$ and $\Delta I$ between each two of the discussed simulations in the paper (See both Figures attached).

**19**

Section 3: Check the use of z, $z_h$, $k_h$, and $k_*$. There appears to be some inconstancy that may lead to confusion.

Here, $z_h$ corresponds to the hub height of 100 m and $k_h$ to the grid point index at hub height whereas $k_*$ to the grid point index either on $z = 75$ m or 125 m. We have a typo in the caption of Fig. 4 (old manuscript version) (now Fig. 5), it should be $z = 125$ m not $z_h$. We changed it.

**20**

Pg 9, Lines 3-5: Would be useful to show plots of TKE to support the point about enhanced production effects on wake recovery.

See Nr. 18.

**21**

Pg 9, Line 10: Check NV_VR, should be NV_NR.

We changed it.

**22**

Figs 4 and 5: Quantification of the y-direction momentum budget could be used to support the assertion that the inflow veer and direction of wake rotation either partially cancel out or enhance wake deflection in the upper and lower portions of the wake.

We agree and added it in the results chapter as: 'This results from a weakening of the wake deflection angle in case of a clockwise rotating wind turbine and an amplification in case of a counterclockwise rotating wind turbine.'

**23**

Are there any data to support the simulation results of the wake you observe in the y-z plane?

We added a comparison to measurements: '...is similar to those observed the simulations of Abkar et al. (2016) (Fig. 16), Vollmer et al. (2017) (Fig. 9), and Englberger and Dörnbrack (2018) (Fig. 6) as well as in measurements of Bodini et al. (2017) (Fig. 12) and Brugger et al. (2019) (Fig. 4).'

[Figure]

**Figure 2.** Vertical and spanwise profiles of $\Delta VD$ in (a), (c), (e), (g) and $\Delta I$ in (b), (d), (f), (h) at $10\,\mathrm{D}$. The spanwise profiles are plotted at 75 m (orange frame, (g), (h)), 100 m (hub height) (green frame, (e) (f)), and 125 m (purple frame (c), (d)). In panels (a) and (b), the coloured lines indicate the altitudes analysed in (c) - (h). Considering a comparison of two simulations A and B (see legend: A vs B), $\Delta VD$ is calculated as the difference between simulation B and simulation A $VD(B) - VD(A)$ and $\Delta I$ as the difference between simulation A and simulation B $I(A) - I(B)$. Therefore, $\Delta VD > 0$ and likewise $\Delta I > 0$ represent a more rapid wake recovery of simulation A in comparison to simulation B, related to a higher turbulence intensity level.

[Figure]

**Figure 3.** Vertical and spanwise profiles of $\Delta VD$ in (a), (c), (e), (g) and $\Delta I$ in (b), (d), (f), (h) at $10\,\mathrm{D}$. The spanwise profiles are plotted at 75 m (orange frame, (g), (h)), 100 m (hub height) (green frame, (e) (f)), and 125 m (purple frame (c), (d)). In panels (a) and (b), the coloured lines indicate the altitudes analysed in (c) - (h). Considering a comparison of two simulations A and B (see legend: A vs B), $\Delta VD$ is calculated as the difference between simulation B and simulation A $VD(B) - VD(A)$ and $\Delta I$ as the difference between simulation A and simulation B $I(A) - I(B)$. Therefore, $\Delta VD > 0$ and likewise $\Delta I > 0$ represent a more rapid wake recovery of simulation A in comparison to simulation B, related to a higher turbulence intensity level.

Fig. 7: Please explain the relatively stronger CW rotation in subfigure 7g compared to 7d and 7h. Why does the wake rotation switch from CCW sense to CW?

This is the fundamental heart of the paper. It is stated as short summary at the end of section 5: 'Consequently, the rotational direction of the flow in the near wake is determined by the rotational direction of the wind turbine, whereas the rotational direction of the flow in the far wake is determined by the ambient wind veer, often dictated by the direction of the Ekman spiral and thus by the sign of the Coriolis force and the hemisphere. If the rotational directions of the flow in the near wake and the boundary-layer flow intensify each other, the same rotational direction persists in the whole wake, as it is the case for a counterclockwise rotating wind turbine (V_CW). Otherwise, the rotational direction of the flow will change in the wake as for a clockwise rotating wind turbine under veered inflow conditions at night in the Northern Hemisphere (V_CCW).'

To make it more clear, we now added a short and informative explanation in the Conclusion: 'The LESs are controlled by the rotational direction of the wind-turbine wake imposed by the blades (clockwise (CW) vs. counterclockwise (CCW) vs. no rotation (NR)) and the inflow wind profiles (wind veer (V) vs. no wind veer (NV)). In detail, in case of a non-rotating actuator, the rotation of the flow in the wake is only determined by the SBL regime. If the rotational direction of the wake and the SBL are the same, also one rotational direction persists in the whole wake, as it would be the case for counterclockwise rotating blades in the Northern Hemisphere (V_CW). In case of the common clockwise rotating blades (V_CCW), the rotational direction in the near wake, imposed by the rotor, differs from the rotational direction in the far wake, induced by the veering wind in the stably stratified regime in the Northern Hemisphere.'

You might consider quantifying the Circulation within the rotor region to quantitatively compare the cases.

We have thought about it. The aim of the paper is to state the difference in the flow field of the wake by changing the rotational direction. It does not present a specific case/location. Therefore, to our opinion, a quantification of the circulation is not mandatory, however we agree with the reviewer, it is essential if specific cases are simulated and compared and we will take it into account for future work.

Fig. 8: Please comment on the relatively strong counter rotating structures observed between the rotor and the corners of the domain. Might these be weaker if the domain were enlarged?

The structures of rotation surrounding the disc are the same for all three cases in Fig. 8. The upper left and lower right corner rotate clockwise while the upper right and the lower left rotate counterclockwise. This persists (at least) up to 7D in all three cases. In case of a non-rotating rotor (NV_NR), the pattern is rather symmetric to the nacelle, whereas the asymmetry and intensification of the 12-3 o'clock and 6-9 o'clock sectors in NV_CCW and the 9-12 o'clock and 3-6 o'clock sectors in NV_CW result from the corresponding rotational direction of the two cases. Due to continuity equation, counter-rotating

vortices are expected. Further, the vorticity of these vortices is rater small in comparison to the main vortex induced by the actuator.

It could be a domain size effect, we cannot test it as the applied NBL precursor simulation used in the parametrization is limited to this spanwise domain. However, the effect only occurs in a SBL simulation with $v = 0$ $NV$, which is a rather artificial situation, whereas it is not the case in the important simulations with veering wind $V$. Further, due to the relatively small vorticity of this vortices it should not be really important for the results of $NV$ and it is not important at all in the veering simulations $V$, representing the new insight of this study.

How might this affect the comparison with the simple model?

Considering the simple model, there is not effect expected. In the non-veered cases, the $v$ and $w$ graphs of Fig. 12 will overlap for the simulation as well as for the model results (plot not shown in paper).

**27**

Sect 4.1, Model Development: It would be useful to relate the models, particularly for the axial component of velocity to existing analytical wake models.

The intention of our simple model is not to be included as computationally fast method in other calculations etc. It was only designed to show that a simple interaction of two physical process is responsible for the completely different behaviour of $v$ and $w$ in the wake. $u$ in the simple model is only of minor importance. Therefore, we do not consider the simple model as analytical wake model. At least not at this stage. However, for future work we will consider it.

For example, what is the physical meaning of the 0.3 used in Eqn. 9?

The physical meaning of the 0.3 factor is explained as: 'We apply a fraction of 0.3, which can be related to $VD_{max} = 0.7$ of the rotating disc simulations in Fig. 3.' It is basically connected the the deceleration of the velocity field behind the disc at about 70% due to the axial force resulting from the wind turbine parametrization, however, not to a deceleration of the streamwise velocity to 0.

It can also be related to the axial induction factor $a$, having the same definition as $VD$. To make to more clear, we added this explanation to make it more clear: 'We apply a fraction of 0.3, which can be related to $VD_{max} = 0.7$ of the rotating disc simulations in Fig. 3 and consequently to an axial induction factor $a$ of 0.7, resulting in a fraction of $0.3 = (1-a)$, as it is the case by calculating the axial force with Momentum Theory Manwell et al. (2002); Hansen (2008).'

What are the axial and tangential induction factors used and are they appropriate for the turbine being modeled?

The model is composed of the ABL interacting with a Rankine Vortex, which is characterized by $R$ and $\omega$. No specific blade characteristics etc. are considered here. Therefore, no comparison of the simple model towards the BEM method is possible. The simple model does not include any turbine characteristics besides $R$ and $\Omega$.

**28**

Be sure to define all variables and subscripts used throughout the model development. For example, _RV, _M, _fad are not explicitly defined.

We added RV to Rankine Vortex: 'The wind-turbine model includes the axial turbine-induced force $\mathbf{F}_x$, as well as the tangential turbine-induced force $\mathbf{F}_\Theta$, which correspond to a Rankine Vortex $(RV)$'

We also added M to the simplified wake model: 'From a linear superposition results for the developed simplified wake model $(M)$: '

_fad was described as: 'The downstream erosion of $v_{WT}$ and $w_{WT}$ with $x_{fad}$ corresponding to a prescribed decay distance at which the atmospheric boundary-layer flow determines the rotational direction of the wake ...'

**29**

Eqns 18 – 20, Given the premise that at least some of the flow conditions are caused by thermal stability, how could stability be included in the model, as the normal and shear components of turbulence fluxes will significantly affect the wake evolution.

This is an interesting topic, however, it would request basically a new model. As it does not contribute to the rotational direction of the wake, we do not consider it in this work.

**30**

How are the values of the parameters x_rec, x_fad, gamma, and delta chosen? Is it possible to estimate them without the flow data from simulations?

The values of $x_{rec}$ and $x_{fad}$ are obtained from the simulation results (see Fig. 12 left column). The magnitude of $\gamma$ and $\delta$ result from archiving the best possible agreement between the model results and the simulation results. We added it as: 'The values of $\gamma$ and $\delta$ are determined by empirical fitting.' We consider empirical fitting appropriate, as the values of $\delta$ and $\gamma$ only modify the amount of $v$ and $w$ downwind, however, not the sign of both.

**31**

Pg 16, Line 14: Please check the single sentence paragraph.

We agree and combined it the the following paragraph describing the situation in detail.

**32**

Fig 9: It would be easier to compare the model with the LES output if they were plotted together and instead separate the CCW and CW cases. Note that the legend above the figure does not coincide with the figure description below for the vertical velocity plots.

We agree, this way of plotting would make it more easy to compare the difference between simulation and simplified model.

Our intent with this figure is to show the rotational direction impact in the simulations and especially in the model. We think this will be more transparent by plotting the simulation results and the simplified model results separately.

We changed the legend.

**33**

Please check that average velocity is defined and used consistently. Note in Fig 9, an overbar is used, but not in the figure description or elsewhere.

We changed in the legend $u$, $v$, and $w$ to $\overline{u}$, $\overline{v}$, and $\overline{w}$.

Are the plotted data based on the 60 deg or 90 deg sectors defined in Table 2?

All data correspond to the 60° averages see 2.3: 'As the features are more distinctive for 60° and 25 m $\leq$ r $\leq$ 50 m sectors, these values are applied for the derivation of the sector characteristics in the following.' See comment 15.

**34**

Consider comparing the results to field data. In particular, there are a number of case experiments that have used lidar to measure the wake. Even if not the data are not directly compatible due to different turbine models or operating conditions, it would be useful to see if measurements see the same trends presented in this study.

We compared specific features (skewness) of the wake structure arising in the SBL in combination with wind veer to measurements: Bodini et al. (2017), Brugger et al. (2019). We can only compare the clockwise rotating blade simulations, therefore no comparison regarding the impact of the rotational direction is possible with field measurements.

**35**

Section 5: It is not clear if thermal stability was actually modeled, or only the Ekman spiral effect on the wind direction. The language used throughout the manuscript may contribute to confusion about what physics are considered in this study. This can be avoided by using explicitly clear language.

We added the definition of $\Theta$, see 12. Further in Section 5, we added 'a lapse rate of 3 K / 200 m' in the text to make it more clear that our simulated stable boundary layer consists of a lapse rate which is combined with a veering wind representing the Ekman spiral.

**36**

Statements in the Conclusion that provide interpretation with various supporting references should be moved up to the Discussion. The Conclusion should focus on the main outcomes of the present study.

This is a matter of style. As the conclusion is relatively brief we think the comparison with other work is appropriate in this conclusion. Further, we refer to Schimel (2012) for writing the conclusion.

**References**

Abkar, M. and Porté-Agel, F.: The effect of atmospheric stability on wind-turbine wakes: A large-eddy simulation study, in: Journal of Physics: Conference Series, vol. 524, p. 012138, IOP Publishing, https://doi.org/10.1088/1742-6596/524/1/012138, 2014.

Abkar, M., Sharifi, A., and Porté-Agel, F.: Wake flow in a wind farm during a diurnal cycle, Journal of Turbulence, 17, 420–441, https://doi.org/10.1080/14685248.2015.1127379, 2016.

Aitken, M. L., Kosović, B., Mirocha, J. D., and Lundquist, J. K.: Large eddy simulation of wind turbine wake dynamics in the stable boundary layer using the Weather Research and Forecasting Model, J Renew Sust Energy, 6, 1529–1539, https://doi.org/10.1063/1.4885111, 2014.

Bak, C., Zahle, F., Bitsche, R., Kim, T., Yde, A., Henriksen, L. C., Hansen, M. H., Blasques, J. P. A. A., Gaunaa, M., and Natarajan, A.: The DTU 10-MW reference wind turbine, in: Danish Wind Power Research 2013, 2013.

Bhaganagar, K. and Debnath, M.: Implications of Stably Stratified Atmospheric Boundary Layer Turbulence on the Near-Wake Structure of Wind Turbines, Energies, 7, 5740–5763, https://doi.org/10.3390/en7095740, 2014.

Bhaganagar, K. and Debnath, M.: The effects of mean atmospheric forcings of the stable atmospheric boundary layer on wind turbine wake, Journal of Renewable and Sustainable Energy, 7, 013 124, https://doi.org/10.1063/1.4907687, 2015.

Bodini, N., Zardi, D., and Lundquist, J. K.: Three-dimensional structure of wind turbine wakes as measured by scanning lidar, Atmospheric Measurement Techniques, 10, 2017.

Brugger, P., Fuertes, F. C., Vahidzadeh, M., Markfort, C. D., and Porté-Agel, F.: Characterization of Wind Turbine Wakes with Nacelle-Mounted Doppler LiDARs and Model Validation in the Presence of Wind Veer, Remote Sensing, 11, 2247, 2019.

Dörenkämper, M., Witha, B., Steinfeld, G., Heinemann, D., and Kühn, M.: The impact of stable atmospheric boundary layers on wind-turbine wakes within offshore wind farms, Journal of Wind Engineering and Industrial Aerodynamics, 144, 146–153, https://doi.org/10.1016/j.jweia.2014.12.011, 2015.

Englberger, A. and Dörnbrack, A.: Impact of Neutral Boundary-Layer Turbulence on Wind-Turbine Wakes: A Numerical Modelling Study, Boundary-Layer Meteorology, 162, 427–449, https://doi.org/10.1007/s10546-016-0208-z, 2017.

Englberger, A. and Dörnbrack, A.: Impact of the diurnal cycle of the atmospheric boundary layer on wind-turbine wakes: a numerical modelling study, Boundary-layer meteorology, 166, 423–448, https://doi.org/10.1007/s10546-017-0309-3, 2018.

Hansen, M. O.: Aerodynamics of wind turbines, vol. 2, Earthscan, London and Sterling, UK and USA, 181 pp, 2008.

Manwell, J., McGowan, J., and Roger, A.: Wind Energy Explained: Theory, Design and Application, Wiley: New York, NY, USA, 134 pp, 2002.

Mirocha, J. D., Kosović, B., Aitken, M. L., and Lundquist, J. K.: Implementation of a generalized actuator disk wind turbine model into the weather research and forecasting model for large-eddy simulation applications, J Renew Sust Energy, 6, 013 104, https://doi.org/10.1063/1.4861061, 2014.

Schimel, J.: Writing science: how to write papers that get cited and proposals that get funded, OUP USA, 2012.

Schmidt, H. and Schumann, U.: Coherent structure of the convective boundary layer derived from large-eddy simulations, J Fluid Mech, 200, 511–562, https://doi.org/10.1017/S0022112089000753, 1989.

Vollmer, L., Lee, J. C., Steinfeld, G., and Lundquist, J.: A wind turbine wake in changing atmospheric conditions: LES and lidar measurements, in: Journal of Physics: Conference Series, vol. 854, p. 012050, IOP Publishing, https://doi.org/10.1088/1742-6596/75/1/012003, 2017.

---

## Author Response (AR2)

**Response letter for the manuscript: Does the rotational direction of a wind turbine impact the wake in a stably stratified atmospheric boundary layer?**

Comments to the Author:

Dear Authors,

As you can see, both reviewers remain very critical of you paper, and expressed the feeling that you were only superficially responding to their concerns. I believe that Reviewer 1 is raising serious concerns regarding your inflow profiles (e.g. Eq 4 and 5 do not correspond to the analytical model proposed by Shapiro & Fedorovich), and also the turbulence generation method that you use can have a significant influence on conclusions. You solve this by improving your boundary conditions and/or by providing evidence (with experimental data, as reviewer 2 suggest), that your results make sense. Also the second comment by Reviewer 1 is to the point. The model that you propose in Section 4 looks very heuristic, and it is not clear why you do not build on existing wake models, etc. I would encourage you to very carefully consider the reviewers comments, as I believe these are make-or-break issues for this work.

Sincerely,

Johan Meyers

Dear Prof. Johan Meyers,

thank you for the time you have devoted to the thoughtful consideration of our manuscript. Following your comments and the reviewers, we carefully considered the criticism and the way that we presented this work. Our intent for the paper is to make people aware that a wind turbine's rotational direction has an impact on the wake under veering inflow. Due to the presentation of some matters in the previous manuscript version, there were a few misunderstandings. Therefore, we have implemented several changes in the manuscript:

- There was a misunderstanding with the 'Simplified Wake Model'. Reviewer 1 suggested a comparison to existing wake models, which make us realize that the name 'Simplified Wake Model' was a perhaps misleading choice. Sect. 4 in the previous version was not intended to be a new wake model, comparable to existing well defined ones. It is simply a device to help to explain why a significant difference in the meridional wake component occurs depending on the rotational direction of the rotor under

veering inflow conditions.

Therefore, in the revised version of the manuscript we have changed the description and included this presentation and discussion in a very simplified equation in Sect. 3 'Analysis of a rotating system under veering inflow'. We further simplified the rather complex equations in the old version of the manuscript, to one analytic equation (Eq. 17). There is no need to use existing wake models as proposed by the reviewer 1, as we are not proposing a model to be used for other purposes like wake steering, energy production estimates, etc. Rather, we present this discussion to highlight the interaction of two physical processes (wake rotation and inflow veer) only. We subsequently interpret our LES results in light of this model of the interaction of these two physical processes.

- Reviewer 1 was concerned about our turbulence generation method. Therefore we have reverted to using established and published wind and turbulence profiles from LES from a diurnal cycle. We extract non-veered inflow from evening boundary layer (EBL) and veered inflow from stable boundary layer (SBL) portions of published in Boundary-Layer Meteorology, editor Fedorovich (Englberger and Dörnbrack, 2018). We therefore no longer reply on artificial inflow profiles (Eq. 4, and 5 in the previous version). The non-veered EBL and the veered SBL wind-turbine simulations are performed for a clockwise (CCW), a counterclockwise (CW), and no (NR) rotor rotation. To further justify our inflow, we compare these inflow profiles resulting from the diurnal cycle simulation to profiles of the Ekman-spiral in the new Eq. 7 and 8. The results show good for a typical SBL value of the eddy viscosity coefficient $\kappa = 0.05$ m$^2$ s$^{-1}$.

These six simulations (SBL_CR, SBL_NR, SBL_CCR, EBL_CR, EBL_NR, EBL_CCR) replace the six existing ones (V_CCW, V_NR, V_CW, NV_CCW, NV_NR, NV_CW) in the previous manuscript version. They have very similar characteristics (wind profiles, potential temperature, turbulence intensity), resulting therefore in similar wake characteristics. Our primary conclusion from the original manuscript persists, that in veered flow, CW rotation and CCW rotation exhibit very different wake characteristics. These results are now presented in Sect. 4: 'A rotating wind-turbine rotor under veering inflow' (Note we changed from CCW (conterclockwise wake) to CR (clockwise rotor), as the think it is more intuitive to give the rotor rotation.) Further, the new SBL simulations apply veer limited to the lower rotor half with a larger directional shear in this region than the 'V' simulations in the previous version.)

- Reviewer 2 suggested that we include a qualitative investigation of $u$ and turbulence intensity $TI$. This new discussion appears in Sect. 4 as well as in Figs. 11 and 12.

- A 'Comparison of a rotating wind turbine under veering inflow to analysis predictions' follows in Sect. 5. It interprets the results in Sect. 4 using the new simple analysis in Sect. 3. This new Section has the same intent as Sect. 4.2 in the previous version.

- Reviewer 2 further suggests a quantitative comparison to measurements. However, the LES result from an idealized diurnal cycle simulation. Therefore, no measurements are available for exactly these conditions. However, in the text, we compare the wake structures (skewness under veering inflow etc.) to other studies, to show that the results are reasonable and consistent with previous work, and benefit from the explanation afforded by the simple analysis in the new Sect. 3. Further, the new set of simulations are based on previously published results and so should not incur any questions regarding the turbulence generation method.

- Despite these extensive changes to the simulations, the simple analysis, and the discussion of the results, the basic motivation of the paper is the same. Therefore, the introduction and conclusion have only minor changes.

In summary, we have put substantial effort into responding to the reviewers' comments and hope you find the manuscript suitable for publication. We have removed the reasons for reviewers' concerns with inflow by using established and published simulations. We have avoided the need for using existing wake models by instead presenting a simplified analytic model that focuses on the two main physical processes at play: inflow veer and wake rotation. Our fundamental point, that turbine rotation interacts with atmospheric profiles in interesting ways, should be much more clear now. The main results have not changed although the mechanism by which we calculated them has changed in response to reviewer suggestions.

Thank you for your consideration.

Reviewer 1: Mayor revisions

The authors did not address the main issues raised in the first review. Also, considering the new information provided in the revised manuscript, unfortunately, I cannot recommend the manuscript to be published. The major comments are:

We thank the reviewer for the time they devoted to considering our manuscript. We have made substancial changes based on their comments, and we hope the manuscript is now suitable for publication.

We carefully considered the criticism and the way that we presented this work. Our intent for the paper is to make people aware that a wind turbine's rotational direction has an impact on the wake under veering inflow. Therefore, we perform extensive changes to the simulations, the simple analysis, and the discussion of the results. Our primary conclusion from the original manuscript persists, that in veered flow, CW rotation and CCW rotation exhibit very different wake characteristics. As the basic motivation of the paper is the same, the introduction and conclusion have only minor changes.

Reviewers comments:

1. Regarding the Simulation: in the revised version and the response letter, the authors provided more details about the inflow conditions. The main issue with this study is that the inflow conditions for veering and non-veering cases are not physical. In other words, the inflow profiles for velocity and Reynolds-stresses for the veering inflow is not similar to (even) idealizes boundary layers with Coriolis effects. The authors can refer to the previous studies on this topic in which a precursor technique is used to generate appropriate inflow conditions for different veering scenarios (e.g., KU Leuven, NREL, EPFL, ...). As the inflow conditions are incorrect and non-physical, the rest of the analysis and the conclusions cannot be justified.

Reviewer 1 was concerned about our turbulence generation method. Therefore we have reverted to using established and published wind and turbulence profiles from LES from a diurnal cycle. We extract non-veered inflow from evening boundary layer (EBL) and veered inflow from stable boundary layer (SBL) portions of published in Boundary-Layer Meteorology, editor Fedorovich (Englberger and Dörnbrack, 2018). We therefore no longer reply on artificial inflow profiles (Eq. 4, and 5 in the previous version). The non-veered EBL and the veered SBL wind-turbine simulations are performed for a clockwise (CCW), a counterclockwise (CW), and no (NR) rotor rotation. To further justify our inflow, we compare these inflow profiles resulting from the diurnal cycle simulation to profiles of the Ekman-spiral in the new Eq. 7 and 8. The results show good for a typical SBL value of the eddy viscosity coefficient $\kappa = 0.05$ m$^2$ s$^{-1}$. We also point out three different observational studies, from different locations, where such inflow veer has been observed.

These six simulations (SBL_CR, SBL_NR, SBL_CCR, EBL_CR, EBL_NR, EBL_CCR) replace the six existing ones (V_CCW, V_NR, V_CW, NV_CCW, NV_NR, NV_CW) in the previous manuscript version. They have very similar characteristics (wind profiles, potential temperature, turbulence intensity), resulting therefore in similar wake characteristics. Our primary conclusion from the original manuscript persists, that in veered flow, CW rotation and CCW rotation exhibit very different wake characteristics. These results are now presented in Sect. 4: 'A rotating wind-turbine rotor under veering inflow' (Note we changed from CCW (conterclockwise wake) to CR (clockwise rotor), as the think it is more intuitive to give the rotor rotation). Further, the new SBL simulations apply veer limited to the lower rotor half with a larger directional shear in this region than the 'V' simulations in the previous version.)

2. Regarding the modeling part: the authors mentioned that "considering complex entrainment processes, we think the mass has not to be conserved at specific downstream regions, as it is the case over the whole domain"! This statement is not true. The presented model by the authors does not conserve mass and momentum. Therefore, it is incorrect. There are already several existing models in the literature that conserve both mass and momentum conservation, and the authors can refer to those.

There was a misunderstanding with the 'Simplified Wake Model'. Reviewer 1 suggested a comparison to existing wake models, which make us realize that the name 'Simplified Wake Model' was a perhaps misleading choice. Sect. 4 in the previous version was not intended to be a new wake model, comparable to existing well defined ones. It is simply a device to help to explain why a significant difference in the meridional wake component occurs depending on the rotational direction of the rotor under veering inflow conditions.

Therefore, in the revised version of the manuscript we have changed the description and included this presentation and discussion in a very simplified equation in Sect. 3 'Analysis of a rotating system under veering inflow'. We further simplified the rather complex equations in the old version of the manuscript, to one analytic equation (Eq. 17). There is no need to use existing wake models as proposed by the reviewer 1, as we are not proposing a model to be used for other purposes like wake steering, energy production estimates, etc. Rather, we present this discussion to highlight the interaction of two physical processes (wake rotation and inflow veer) only. We subsequently interpret our LES results in light of this model of the interaction of these two physical processes.

Reviewer 2: Minor revisions

The authors have provided satisfactory response to my comments and I thank them for taking the time to expand on their presentation and clarify a number of items.

We thank the reviewer for the considerable time they have devoted to helping us improve the presentation of our work.

We carefully considered the criticism and the way that we presented this work. Our intent for the paper is to make people aware that a wind turbine's rotational direction has an impact on the wake under veering inflow. Therefore, we perform extensive changes to the simulations, the simple analysis, and the discussion of the results. Our primary conclusion from the original manuscript persists, that in veered flow, CW rotation and CCW rotation exhibit very different wake characteristics. As the basic motivation of the paper is the same, the introduction and conclusion have only minor changes.

A few general comments to the revised version of the manuscript:

- Reviewer 1 was concerned about our turbulence generation method. Therefore we have reverted to using established and published wind and turbulence profiles from LES from a diurnal cycle. We extract non-veered inflow from evening boundary layer (EBL) and veered inflow from stable boundary layer (SBL) portions of published in Boundary-Layer Meteorology, editor Fedorovich (Englberger and Dörnbrack, 2018). We therefore no longer reply on artificial inflow profiles (Eq. 4, and 5 in the previous version). The non-veered EBL and the veered SBL wind-turbine simulations are performed for a clockwise (CCW), a counterclockwise (CW), and no (NR) rotor rotation. To further justify our inflow, we compare these inflow profiles resulting from the diurnal cycle simulation to profiles of the Ekman-spiral in the new Eq. 7 and 8. The results show good for a typical SBL value of the eddy viscosity coefficient $\kappa = 0.05$ m$^2$ s$^{-1}$.

  These six simulations (SBL_CR, SBL_NR, SBL_CCR, EBL_CR, EBL_NR, EBL_CCR) replace the six existing ones (V_CCW, V_NR, V_CW, NV_CCW, NV_NR, NV_CW) in the previous manuscript version. They have very similar characteristics (wind profiles, potential temperature, turbulence intensity), resulting therefore in similar wake characteristics. Our primary conclusion from the original manuscript persists, that in veered flow, CW rotation and CCW rotation exhibit very different wake characteristics. These results are now presented in Sect. 4: 'A rotating wind-turbine rotor under veering inflow' (Note we changed from CCW (conterclockwise wake) to CR (clockwise rotor), as the think it is more intuitive to give the rotor rotation.) Further, the new SBL simulations apply veer limited to the lower rotor half with a larger directional shear in this region than the 'V' simulations in the previous version.)

Reviewers comments:

I would still like to see quantitative comparison to measurements. The paper reads as, ...trust the simulations, they are believed to represent the relavent physics correctly. Even if data isn't available to compare with every case that was considered, that's fine. That is the power of simulation to extend observation to investigate a broader possible conditions.

Reviewer 2 suggests a quantitative comparison to measurements. However, the LES result from an idealized diurnal cycle simulation. Therefore, no measurements are available for exactly these conditions. However, in the text, we compare the wake structures (skewness under veering inflow etc.) to other studies, to show that the results are reasonable and consistent with previous work, and benefit from the explanation afforded by the simple analysis in the new Sect. 3. Further, the new set of simulations are based on previously published results and so should not incur any questions regarding the turbulence generation method.

What is still lacking in this study is to leverage the advantage of the power of LES to provide information about unsteady phenomena. Otherwise, why do you need LES and not simply run RANS to show the relative behavior.

There was a misunderstanding with the 'Simplified Wake Model', which make us realize that the name 'Simplified Wake Model' was a perhaps misleading choice. Sect. 4 in the previous version was not intended to be a new wake model, comparable to existing well defined ones. It is simply a device to help to explain why a significant difference in the meridional wake component occurs depending on the rotational direction of the rotor under veering inflow conditions, as simplification, for a laminar inflow without any turbulence.

Therefore, in the revised version of the manuscript we have changed the description and included this presentation and discussion in a very simplified equation in Sect. 3 'Analysis of a rotating system under veering inflow'. We further simplified the rather complex equations in the old version of the manuscript, to one analytic equation (Eq. 17). This should highlight the laminar inflow without any turbulence and interaction between the three wind components considered.

We present this analysis to highlight the interaction of two physical processes (wake rotation and inflow veer). We subsequently interpret our LES results in light of this model of the interaction of these two physical processes. Comparing the LES results (with turbulent flow) to the simple analysis (for laminar flow), we highlight the important impact of turbulence and the benefit of using LES for this study.

An example: In the LES of the SBL WT simulations, veer is limited to the lower rotor part. In the analysis (laminar inflow), there is not impact in the upper rotor part in this case, however, we see an impact in the LES SBL WT simulations also in the upper rotor part, which is an effect of the turbulence which is considered in LES.

The addition of the comparison of velocity deficit and total turbulence intensity is helpful. It would be helpful to also include the comparison of velocity

profiles or deficit profiles and turbulence intensity or added turbulence intensity at a couple of wake positions in the vertical and horizontal planes to get a better feel for how the different cases compare. I think this will also help readers interpret the profiles of difference, which are a bit noisy and hard to read.

Reviewer 2 suggested that we include a qualitative investigation of $u$ and turbulence intensity $TI$. This new discussion appears in Sect. 4 as well as in Fig. 11 and 12.

Overall, this is an interesting comparison of cases and clearly a lot of work went into the simulations.
Thank you!

**Does the rotational direction of a wind turbine impact the wake in a stably stratified atmospheric boundary layer?**

Antonia Englberger[1], Andreas Dörnbrack[1], and Julie K. Lundquist[2,3]

[1]German Aerospace Center, Institute of Atmospheric Physics, Oberpfaffenhofen, Germany
[2]Department of Atmospheric and Oceanic Sciences, University of Colorado Boulder, Boulder, USA
[3]National Renewable Energy Laboratory, Golden, Colorado, USA

**Correspondence:** Antonia Englberger (antonia.englberger@dlr.de)

**Abstract.** Stably stratified atmospheric boundary layers are often characterized by a veering wind profile, in which the wind direction changes clockwise with height in the Northern Hemisphere. Wind-turbine wakes respond to this veer in the incoming wind by stretching from a circular shape into an ellipsoid. We investigate the relationship between this stretching and the direction of the turbine rotation by means of large-eddy simulations. Clockwise rotating, counterclockwise rotating, and non-rotating actuator disc turbines are embedded in wind fields  with no wind veer  and in wind fields with  a Northern Hemispheric Ekman spiral, resulting in six combinations of rotor rotation and inflow wind condition. The  wake strength, extension, width and deflection depend on the interaction of the meridional component of Ekman spiral with the rotational direction of the actuator disc, whereas the direction of the disc rotation  only marginally modifies the wake if no veer is present. The differences result from the  amplification or weakening/reversion of the spanwise and the vertical wind components  due to the effect of the superposed disc rotation. They are also present in the streamwise wind component of the wake and in the total turbulence intensity. In the case of a counterclockwise rotating actuator disc, the  spanwise and vertical wind components increase directly behind the rotor resulting in the same rotational direction in the whole wake  while its strength decreases downwind. In the case of a clockwise rotating actuator disc, however, the  spanwise and vertical wind components of the near wake are weakened or even reversed in comparison to the inflow. This weakening/reversion results in a downwind increase of the strength of the flow rotation in the wake or even a different rotational direction in the near wake in comparison to the far wake. The physical mechanism responsible for this difference  can be explained by a simple linear superposition of  a veering inflow with a Rankine vortex.

*Copyright statement.* The copyright of the authors Antonia Englberger and Andreas Dörnbrack for this publication are transferred to Deutsches Zentrum für Luft- und Raumfahrt e. V., the German Aerospace Center. 
[revised manuscript text omitted]